# LEARNING-AUGMENTED $k$-MEANS CLUSTERING

**Jon C. Ergun**
Georgetown Day School
jergun22@gds.org

**Zhili Feng**
Carnegie Mellon University
zhilif@andrew.cmu.edu

**Sandeep Silwal**
MIT
silwal@mit.edu

**David P. Woodruff**
Carnegie Mellon University
dwoodruf@andrew.cmu.edu

**Samson Zhou**
Carnegie Mellon University
samsonzhou@gmail.com

## ABSTRACT

$k$-means clustering is a well-studied problem due to its wide applicability. Unfortunately, there exist strong theoretical limits on the performance of any algorithm for the $k$-means problem on worst-case inputs. To overcome this barrier, we consider a scenario where "advice" is provided to help perform clustering. Specifically, we consider the $k$-means problem augmented with a predictor that, given any point, returns its cluster label in an approximately optimal clustering up to some, possibly adversarial, error. We present an algorithm whose performance improves along with the accuracy of the predictor, even though naïvely following the accurate predictor can still lead to a high clustering cost. Thus if the predictor is sufficiently accurate, we can retrieve a close to optimal clustering with nearly optimal runtime, breaking known computational barriers for algorithms that do not have access to such advice. We evaluate our algorithms on real datasets and show significant improvements in the quality of clustering.

## 1 INTRODUCTION

Clustering is a fundamental task in data analysis that is typically one of the first methods used to understand the structure of large datasets. The most common formulation of clustering is the $k$-means problem where given a set $P \subset \mathbb{R}^d$ of $n$ points, the goal is to find a set of centers $C \subset \mathbb{R}^d$ of $k$ points to minimize the objective $\text{cost}(P, C) = \sum_{p \in P} \min_{c \in C} \|p - c\|_2^2$. (1)

Despite decades of work, there exist strong theoretical limitations about the performance of any algorithm for the $k$-means problem. Finding the optimal set $C$ is NP-hard even for the case of $k = 2$ (Dasgupta, 2008) and even finding an approximate solution with objective value that is within a factor $1.07$ of the optimal solution is NP-hard (Cohen-Addad & S., 2019; Lee et al., 2017). Furthermore, the best-known practical *polynomial* time algorithms can only provably achieve a large constant factor approximation to the optimal clustering, e.g., the 50-approximation in Song & Rajasekaran (2010), or use techniques such as linear programming that do not scale, e.g., the 6.357-approximation in Ahmadian et al. (2020).

A natural approach to overcome these computational barriers is to leverage the fact that in many applications, the input is often not arbitrary and contains auxiliary information that can be used to construct a good clustering, e.g., in many applications, the input can be similar to past instances. Thus, it is reasonable to create a (possibly erroneous) predictor by using auxiliary information or through clusterings of similar datasets, which can inform the proper label of an item in our current dataset. Indeed, inspired by the developments in machine learning, many recent papers have studied algorithms augmented with predictions (Mitzenmacher & Vassilvitskii, 2020). Such algorithms utilize a predictor that, when invoked, provides an (imperfect) prediction for future inputs. The predictions are then used by the algorithm to improve performance (see references in Section 1.3).

Hence, we consider the problem of $k$-means clustering given additional access to a predictor that outputs advice for which points should be clustered together, by outputting a label for each point. The goal is to find $k$ centers that minimize objective (1) and assign each point to one of these centers.

The question is then whether one can utilize such predictions to boost the accuracy and runtime of clustering of new datasets. Our results demonstrate the answer in the affirmative.

**Formal learning-augmented problem definition.** Given a set $P \subseteq \mathbb{R}^d$ of $n$ points, the goal is to find a set of $k$ points $C$ (called centers) to minimize objective (1). In the learning-augmented setting, we assume we have access to a predictor $\Pi$ that provides information about the label of each point consistent with a $(1 + \alpha)$-approximately optimal clustering $\mathcal{C}$. We say that a predictor has *label error rate* $\lambda \leq \alpha$ if for each label $i \in [k] := \{1, \ldots, k\}$, $\Pi$ errs on at most a $\lambda \leq \alpha$ fraction of all points in cluster $i$ in $\mathcal{C}$, and $\Pi$ errs on at most a $\lambda \leq \alpha$ fraction of all points given label $i$ by $\Pi$. In other words, $\Pi$ has at least $(1 - \lambda)$ precision and recall for each label.

Our predictor model subsumes both random and adversarial errors by the predictor. For example if the cluster sizes are somewhat well-balanced, then a special case of our model is when $\Pi(p)$ outputs the correct label of point $p \in P$ with some probability $1 - \lambda$ and otherwise outputs a random label in $[k]$ with probability $\lambda$. The example where the predictor outputs an *adversarial* label instead of a random label with probability $\lambda$ also falls under our model. For more detail, see Theorems 2.1 and 3.4. We also adjust our algorithm to have better performance when the errors are random rather than adversarial in the supplementary material.

## 1.1 MOTIVATION FOR OUR WORK

We first motivate studying $k$-means clustering under the learning-augmented algorithms framework.

**Overcoming theoretical barriers.** As stated above, *no* polynomial time algorithm can achieve better than a constant factor approximation to the optimal clustering. In addition, the best provable approximation guarantees by polynomial time algorithms have a large constant factor (for example the 50 approximation in Song & Rajasekaran (2010)), or use methods which do not scale (such as the linear programming based algorithm in Ahmadian et al. (2020) which gives a 6.357-approximation). Therefore, it is of interest to study whether a natural assumption can overcome these complexity barriers. In our work, we show that knowing the true labels up to some possibly adversarial noise can give us arbitrarily good clusterings, depending on the noise level, which breaks these computational barriers. Furthermore, we present an algorithm that runs in nearly *linear* time, rather than just polynomial time. Lastly, we introduce tools from the robust statistics literature to study $k$-means clustering rather than the distance-based sampling procedure that is commonly analyzed (this is the basis of `kmeans++`). This new toolkit and connection could have further applications in other learning-augmented clustering problems.

**Practical considerations.** In practice, good predictors can be learned for datasets with auxiliary information. For a concrete example, we can take *any* dataset that has a train/test split and use a clustering on the training dataset to help us cluster the testing portion of the dataset. Therefore, datasets do not have to be specifically curated to fit our modelling assumption, which is a requirement in other modelling formulations that leverage extra information such as the SSAC model discussed in Section 1.3. A predictor can also be created from the natural class of datasets that vary over time, such as Census data or spectral clustering for temporal graphs (graphs slowly varying over time). For this class of datasets, a clustering from an earlier time step can function as a predictor for later time steps. Lastly, we can simply use the labels given by another clustering algorithm (such as `kmeans++`) or heuristic as a predictor. Therefore, predictors are readily and easily available for a wide class of natural datasets.

**Following the predictor alone is insufficient.** Given a predictor that outputs noisy labels, it is conceivable that its output alone can give us a good clustering relative to optimal. However, this is not the case and naïvely using the label provided by the predictor for each point can result in an arbitrarily bad solution, even when the predictor errs with low probability. For example, consider a cluster of $\frac{n}{2}$ points at the origin and a cluster of $\frac{n}{2}$ points at $x = 1$. Then for $k = 2$, choosing centers at the origin and at $x = 1$ induces a $k$-means clustering cost of zero. However, even for a predictor that errs with probability $\frac{1}{n}$, some point will be mislabeled with constant probability, which results in a positive $k$-means clustering cost, and so does not provide a relative error approximation. Thus, using the provided labels by the predictor can induce an arbitrarily bad clustering, even as the label error rate of the predictor tends to zero. This subtlety makes the model rich and interesting, and requires us to create non-trivial clustering algorithms.

**Predictors with adversarial errors.** Since the predictor is separate from the clustering algorithm, interference with the output of the predictor following the clustering algorithm's query can be a source of non-random noise. Thus any scenario in which communication is performed over a noisy channel (for example, if the predictor is hosted at one server and the algorithm is hosted at another server) is susceptible to such errors. Another source of adversarial failure by the predictor is when the predictor is trained on a dataset that can be generated by an adversary, such as in the context of adversarial machine learning. Moreover, our algorithms have better guarantees when the predictor does not fail adversarially, e.g., see the supplementary material).

## 1.2 OUR RESULTS

In this paper we study "learning-augmented" methods for efficient $k$-means clustering. Our contributions are both theoretical and empirical. On the theoretical side, we introduce an algorithm that provably solves the $k$-means problem almost optimally, given access to a predictor that outputs a label for each point $p \in P$ according to a $(1 + \alpha)$-approximately optimal clustering, up to some noise. Specifically, suppose we have access to a predictor $\Pi$ with label error rate $\lambda$ upper bounded by a parameter $\alpha$. Then, Algorithm 1 outputs a set of centers $\widetilde{C}$ in $\tilde{O}(knd)$ time[1], such that $\text{cost}(P, \widetilde{C}) \leq (1 + O(\alpha)) \cdot \text{cost}(P, C^{opt})$, where $C^{opt}$ is an optimal set of centers. We improve the runtime in Section 3 by introducing Algorithm 3, which has the same error guarantees, but uses $\tilde{O}(nd)$ runtime, which is *nearly optimal* since one needs at least $nd$ time to read the points for dense inputs (Theorem 3.4, and Remark A.14).

To output labels for all points, Algorithm 3 requires $n$ queries to the predictor. However, if the goal is to just output centers for each cluster, then we only require $\tilde{O}(k/\alpha)$ queries. This is essentially optimal; we show in Theorem 3.5 that **any** polynomial time algorithm must perform approximately $\widetilde{\Omega}(k/\alpha)$ queries to output a $1 + \alpha$-approximate solution assuming the Exponential Time Hypothesis, a well known complexity-theoretic assumption (Impagliazzo & Paturi, 2001). Note that one could ignore the oracle entirely, but then one is limited by the constant factor hardness for polynomial time algorithms, which we bypass with a small number of queries.

Surprisingly, we **do not** require assumptions that the input is well-separated or approximation-stable (Braverman et al., 2011; Balcan et al., 2013), which are assumed in other works. Finally in the supplementary material, we also give a learning-augmented algorithm for the related problem of *k-median* clustering, which has less algebraic structure than that of $k$-means clustering. We also consider a *deletion* predictor, which either outputs a correct label or a failure symbol $\perp$ and give a $(1 + \alpha)$-approximation algorithm even when the "deletion rate" is $1 - 1/\text{poly}(k)$.

On the empirical side, we evaluate our algorithms on real and synthetic datasets. We experimentally show that good predictors can be learned for all of our varied datasets, which can aid in clustering. We also show our methodology is more robust than other heuristics such as random sampling.

## 1.3 RELATED WORK

**Learning-augmented algorithms.** Our paper adds to the growing body of work on learning-augmented algorithms. In this framework, additional "advice" from a possibly erroneous predictor is used to improve performance of classical algorithms. For example, a common predictor is a "heaviness" predictor that outputs how "important" a given input point is. It has been shown that such predictors can be learned using modern machine learning techniques or other methods on training datasets and can be successfully applied to similar testing datasets. This methodology has found applications in improving data structures (Kraska et al., 2018; Mitzenmacher, 2018), streaming algorithms (Hsu et al., 2019; Jiang et al., 2020), online algorithms (Lykouris & Vassilvtiskii, 2018; Purohit et al., 2018), graph algorithms (Dai et al., 2017), and many other domains (Mousavi et al., 2015; Wang et al., 2016; Bora et al., 2017; Sablayrolles et al., 2019; Dong et al., 2020; Sanchez et al., 2020; Eden et al., 2021). See Mitzenmacher & Vassilvitskii (2020) for an overview and applications.

**Clustering with additional information.** There have been numerous works that study clustering in a semi-supervised setting where extra information is given. Basu et al. (2004) gave an active learning framework of clustering with "must-link"/"cannot-link" constraints, where an algorithm is allowed

---

[1]The notation $\widetilde{O}$ hides logarithmic factors.

to interact with a predictor that determines if two points must or cannot belong to the same cluster. Their objective function is different than that of $k$-means and they do not give theoretical bounds on the quality of their solution. Balcan & Blum (2008) and Awasthi et al. (2017) studied an interactive framework for clustering, where a predictor interactively provides feedback about whether or not to split a current cluster or merge two clusters. Vikram & Dasgupta (2016) also worked with an interactive oracle but for the Bayesian hierarchical clustering problem. These works differ from ours in their assumptions since their predictors must answer different questions about partitions of the input points. In contrast, Howe (2017) used logistic regression to aid $k$-means clustering but do not give any theoretical guarantees.

The framework closest in spirit to ours is the semi-supervised active clustering framework (SSAC) introduced by Ashtiani et al. (2016) and further studied by Kim & Ghosh (2017); Mazumdar & Saha (2017); Gamlath et al. (2018); Ailon et al. (2018); Chien et al. (2018); Huleihel et al. (2019). The goal of this framework is also to produce a $(1 + \alpha)$-approximate clustering while minimizing the number of queries to a predictor that instead answers queries of the form "same-cluster$(u, v)$", which returns 1 if points $u, v \in P$ are in the same cluster in a particular optimal clustering and 0 otherwise. Our work differs from the SSAC framework in terms of both runtime guarantees, techniques used, and model assumptions, as detailed below.

We briefly compare to the most relevant works in the SSAC framework, which are Ailon et al. (2018) and Mazumdar & Saha (2017). First, the runtime of Ailon et al. (2018) is $O(ndk^9/\alpha^4)$ even for a perfectly accurate predictor, while the algorithm of Mazumdar & Saha (2017) uses $O(nk^2)$ queries and runtime $\tilde{O}(ndk^2)$. By comparison, we use significantly fewer queries, with near linear runtime $\tilde{O}(nd)$ even for an erroneous predictor. Moreover, a predictor of Mazumdar & Saha (2017) independently fails each query with probability $p$ so that repeating with pairs containing the same point can determine the correct label of a point whereas our oracle will *always repeatedly fail with the same query*, so that repeated queries do not help.

The SSAC framework uses the predictor to perform importance sampling to obtain a sufficient number of points from each cluster whereas we use techniques from robust mean estimation, dimensionality reduction, and approximate nearest neighbor data structures. Moreover, it is unclear how the SSAC predictor can be implemented in practice to handle adversarial corruptions. One may consider simulating the SSAC predictor using information from individual points by simply checking if the labels of the two input points are the same. However, if a particular input is mislabeled, then all of the pairs containing this input can also be reported incorrectly, which violates their independent noise assumption. Finally, the noisy predictor algorithm in Ailon et al. (2018) invokes a step of recovering a hidden clique in a stochastic block model, making it prohibitively costly to implement.

Lastly, in the SSAC framework, datasets need to be specifically created to fit into their model since one requires pairwise information. In contrast, our predictor requires information about individual points, which can be learned from either a training dataset, from past similar datasets, or from another approximate or heuristic clustering and is able to handle adversarial corruptions. Thus, we obtain significantly faster algorithms while using an arguably more realistic predictor.

**Approximation stability.** Another approach to overcome the NP-hardness of approximation for $k$-means clustering is the assumption that the underlying dataset follows certain distributional properties. Introduced by Balcan et al. (2013), the notion of $(c, \alpha)$-approximate stability (Agarwal et al., 2015; Awasthi et al., 2019; Balcan et al., 2020) requires that every $c$-approximation is $\alpha$-close to the optimal solution in terms of the fraction of incorrectly clustered points. In contrast, we allow inputs so that an arbitrarily small fraction of incorrectly clustered points can induce arbitrarily bad approximations, as previously discussed, e.g., in Section 1.1.

## 2    LEARNING-AUGMENTED $k$-MEANS ALGORITHM

**Preliminaries.** We use $[n]$ to denote the set $\{1, \ldots, n\}$. Given the set of cluster centers $C$, we can partition the input points $P$ into $k$ clusters $\{C_1, \ldots, C_k\}$ according to the closest center to each point. If a point is grouped in $C_i$ in the clustering, we refer to its label as $i$. Note that labels can be arbitrarily permuted as long as the labeling across the points of each cluster is consistent. It is well-known that in $k$-means clustering, the $i$-th center is given by the coordinate-wise mean of the

---

**Algorithm 1** Learning-augmented $k$-means clustering

---

**Input:** A point set $X$ with labels given by a predictor $\Pi$ with label error rate $\lambda$

**Output:** $(1 + O(\alpha))$-approximate $k$-means clustering of $X$

1: **for** $i = 1$ to $i = k$ **do**
2:    Let $Y_i$ be the set of points with label $i$.
3:    Run CRDEST for each of the $d$ coordinates of $Y_i$.

4:    Let $C_i'$ be the coordinate-wise outputs of CRDEST.
5: **end for**
6: **Return** clustering with centers $C_1', \ldots, C_k'$.

---

**Algorithm 2** Coordinate-wise estimation CRDEST

---

**Input:** Points $x_1, \ldots, x_{2m} \in \mathbb{R}$, corruption level $\lambda \leq \alpha$

1: Randomly partition the points into two groups $X_1, X_2$ of size $m$.
2: Let $I = [a, b]$ be the shortest interval containing $m(1 - 5\alpha)$ points of $X_1$.
3: $Z \leftarrow X_2 \cap I$
4: $z \leftarrow \frac{1}{|Z|} \sum_{x \in Z} x$
5: **Return** $z$

---

points in $C_i$. Given $x \in \mathbb{R}^d$ and a set $C \subset \mathbb{R}^d$, we define $d(x, C) = \min_{c \in C} \|x - c\|_2$. Note that there may be many approximately optimal clusterings but we consider a fixed one for our analysis.

## 2.1 OUR ALGORITHM

Our main result is an algorithm for outputting a clustering that achieves a $(1 + 20\alpha)$ approximation [2] to the optimal objective cost when given access to approximations of the correct labeling of the points in $P$. We first present a suboptimal algorithm in Algorithm 1 for intuition and then optimize the runtime in Algorithm 3, which is provided in Section 3.

The intuition for Algorithm 1 is as follows. We first address the problem of identifying an approximate center for each cluster. Let $C_1^{opt}, \cdots, C_k^{opt}$ be an optimal grouping of the points and consider all the points labeled $i$ by our predictor for some fixed $1 \leq i \leq k$. Since our predictor can err, a large number of points that are not in $C_i^{opt}$ may also be labeled $i$. This is especially problematic when points that are "significantly far" from cluster $C_i^{opt}$ are given the label $i$, which may increase the objective function arbitrarily if we simply take the mean of the points labeled $i$ by the predictor.

To filter out such outliers, we consider a two step view from the robust statistics literature, e.g., Prasad et al. (2019); these two steps can respectively be interpreted as a "training" phase and a "testing" phase that removes "bad" outliers. We first randomly partition the points that are given label $i$ into two groups, $X_1$ and $X_2$, of equal size. We then estimate the mean of $C_i^{opt}$ using a coordinate-wise approach through Algorithm 2 (CRDEST), decomposing the total cost as the sum of the costs in each dimension.

For each coordinate, we find the smallest interval $I$ that contains a $(1 - 4\alpha)$ fraction of the points in $X_1$. We show that for label error rate $\lambda \leq \alpha$, this "training" phase removes any outliers and thus provides a rough estimation to the location of the "true" points that are labeled $i$. To remove dependency issues, we then "test" $X_2$ on $I$ by computing the mean of $X_2 \cap I$. This allows us to get empirical centers that are a sufficiently good approximation to the coordinates of the true center for each coordinate. We then repeat on the other labels. The key insight is that the error from mean-estimation can be directly charged to the approximation error due to the special structure of the $k$-means problem. Our main theoretical result considers predictors that err on at most a $\lambda$-fraction of all cluster labels. Note that all omitted proofs appear in the supplementary material.

**Theorem 2.1.** *Let $\alpha \in (10 \log n / \sqrt{n}, 1/7)$, $\Pi$ be a predictor with label error rate $\lambda \leq \alpha$, and $\gamma \geq 1$ a sufficiently large constant. If each cluster in the $(1 + \alpha)$-approximately optimal $k$-means clustering of the predictor has at least $\gamma \eta k / \alpha$ points, then Algorithm 1 can be used to output a $(1 + 20\alpha)$-approximation to the $k$-means objective with prob. $1 - 1/\eta$, using $O(kdn \log n)$ runtime.*

We improve the running time to $O(nd \log n + \text{poly}(k, \log n))$ in Theorem 3.4 in Section 3. Our algorithms can also tolerate similar error rates when failures correspond to random labels, adversarial labels, or a special failure symbol.

---

[2]Note that we have not attempted to optimize the constant 20.

**Error rate $\lambda$ vs. accuracy parameter $\alpha$.** We emphasize that $\lambda$ is the error rate of the predictor and $\alpha$ is only some loose upper bound on $\lambda$. It is reasonable that some algorithms can provide lossy guarantees on their outputs, which translates to the desired loose upper bound $\alpha$ on the accuracy of the predictor. Even if is not known, multiple instances of the algorithm can be run in parallel with separate exponentially decreasing "guesses" for the value $\alpha$. We can simply return the best clustering among these algorithms, which will provide the same theoretical guarantees as if we set $\alpha = 1.01\lambda$, for example. Thus $\alpha$ does not need to be known in advance and it does not need to be tuned as a hyperparameter.

## 3 Nearly Optimal Runtime Algorithm

We now describe Algorithm 3, which is an optimized runtime version of Algorithm 1 and whose guarantees we present in Theorem 3.4. The bottleneck for Algorithm 1 is that after selecting $k$ empirical centers, it must still assign each of the $n$ points to the closest empirical center. The main intuition for Algorithm 3 is that although reading all points uses $O(nd)$ time, we do not need to spend $O(dk)$ time per point to find its closest empirical center, if we set up the correct data structures. In fact, as long as we assign each point to a "relatively good" center, the assigned clustering is still a "good" approximation to the optimal solution. Thus we proceed in a similar manner as before to sample a number of input points and find the optimal $k$ centers for the sampled points.

We use dimensionality reduction and an approximate nearest neighbor (ANN) data structure to efficiently assign each point to a "sufficiently close" center. Namely if a point $p \in P$ should be assigned to its closest empirical $C_i$ then $p$ must be assigned to some empirical center $C_j$ such that $\|p - C_j\|_2 \leq 2\|p - C_i\|_2$. Hence, points that are not assigned to their optimal centers only incur a "small" penalty due to the ANN data structure and so the cost of the clustering does not increase "too much" in expectation. Formally, we need the following definitions.

**Theorem 3.1** (JL transform). *Johnson & Lindenstrauss (1984) Let $d(\cdot, \cdot)$ be the standard Euclidean norm. There exists a family of linear maps $A : \mathbb{R}^d \to \mathbb{R}^k$ and an absolute constant $C > 0$ such that for any $x, y \in \mathbb{R}^d$, $\mathbf{Pr}\left[\phi \in A, d(\phi(x), \phi(y)) \in (1 \pm \alpha)d(x, y)\right] \geq 1 - e^{-C\alpha^2 k}$.*

**Definition 3.2** (Terminal dimension reduction). *Given a set of points called terminals $C \subset \mathbb{R}^d$, we call a map $f : \mathbb{R}^d \to \mathbb{R}^k$ a terminal dimension reduction with distortion $D$ if for every terminal $c \in C$ and point $p \in \mathbb{R}^d$, we have $d(p, c) \leq d(f(p), f(c)) \leq D \cdot d(p, c)$.*

**Definition 3.3** (Approximate nearest neighbor search). *Given a set $P$ of $n$ points in a metric space $(X, d)$, a $(c, r)$-approximate nearest neighbor search (ANN) data structure takes any query point $q \in X$ with non-empty $\{p \in P : 0 < d(p, q) \leq r\}$ and outputs a point in $\{p \in P : 0 < d(p, q) \leq cr\}$.*

To justify the guarantees of Algorithm 3, we need runtime guarantees on creating a suitable dimensionality reduction map and an ANN data structure. These are from Makarychev et al. (2019) and Indyk & Motwani (1998); Har-Peled et al. (2012); Andoni et al. (2018) respectively, and are stated in Theorems $A.12$ and $A.13$ in the supplementary section. They ensure that each point is mapped to a "good" center. Thus, we obtain our main result describing the guarantees of Algorithm 3.

**Theorem 3.4.** *Let $\alpha \in (10 \log n / \sqrt{n}, 1/7)$, $\Pi$ be a predictor with label error rate $\lambda \leq \alpha$, and $\gamma \geq 1$ be a sufficiently large constant. If each cluster in the optimal $k$-means clustering of the predictor has at least $\gamma k \log k / \alpha$ points, then Algorithm 3 outputs a $(1 + 20\alpha)$-approximation to the $k$-means objective with probability at least $3/4$, using $O(nd \log n + \text{poly}(k, \log n))$ total time.*

Note that if we wish to only output the $k$ centers rather than labeling all of the input points, then the query complexity of Algorithm 3 is $\widetilde{O}(k/\alpha)$ (see Step 1 of Algorithm 3) with high probability. We show in the supplementary material that this is nearly optimal.

**Theorem 3.5.** *For any $\delta \in (0, 1]$, any algorithm that makes $O\left(\frac{k^{1-\delta}}{\alpha \log n}\right)$ queries to the predictor with label error rate $\alpha$ cannot output a $(1 + C\alpha)$-approximation to the optimal $k$-means clustering cost in time $2^{O(n^{1-\delta})}$ time, assuming the Exponential Time Hypothesis.*

---

**Algorithm 3** Fast learning-augmented algorithm for $k$-means clustering.

---

**Input:** A point set $X$, a predictor $\Pi$ with label error rate $\lambda \leq \alpha$, and a tradeoff parameter $\zeta$
**Output:** A $(1 + \alpha)$-approximate $k$-means clustering of $X$
 1: Form $S$ by sampling each point of $X$ with probability $\frac{100 \log k}{\alpha |A_x|}$ where $A_x$ is the set of points with the same label as $x$ according to $\Pi$.
 2: Let $C_1, \ldots, C_k$ be the output of Algorithm 1 on $S$.
 3: Let $\phi_2$ be a random JL linear map with distortion $\frac{5}{4}$, i.e., dimension $O(\log n)$.
 4: Let $\phi_1$ be a terminal dimension reduction with distortion $\frac{5}{4}$.
 5: Let $\phi := \phi_1 \circ \phi_2$ be the composition map.
 6: Let $A$ be a $(2, r)$-ANN data structure on the points $\phi(C_1), \ldots, \phi(C_k)$.
 7: **for** $x \in X$ **do**
 8:     Let $\ell_x$ be the label of $x$ from $\Pi$.
 9:     $\varrho \leftarrow d(x, C_{\ell_x})$
10:     Query $A$ to find the closest center $\phi(C_{p_x})$ to $x$ with $r = \frac{\varrho}{2}$.
11:     **if** $d(x, C_{p_x}) < 2d(x, C_{\ell_x})$ **then**
12:         Assign label $p_x$ to $x$.
13:     **else**
14:         Assign label $\ell_x$ to $x$.
15:     **end if**
16: **end for**

---

## 4    EXPERIMENTS

In this section we evaluate Algorithm 1 empirically on real datasets. We choose to implement Algorithm 1, as opposed to the runtime optimal Algorithm 3, for simplicity and because the goal of our experiments is to highlight the error guarantees of our methodology, which both algorithms share. Further, we will see that Algorithm 1 is already very fast compared to alternatives. Thus, we implement the simpler of the two algorithms. We primarily fix the number of clusters to be $k = 10$ and $k = 25$ throughout our experiments for all datasets. Note that our predictors can readily generalize to other values of $k$ but we focus on these two values for clarity. All of our experiments were done on a CPU with i5 2.7 GHz dual core and 8 GB RAM. Furthermore, all our experimental results are averaged over 20 independent trials and $\pm$ one standard deviation error is shaded when applicable. We give the full details of our datasets below.

**1) Oregon**: Dataset of 9 graph snapshots sampled across 3 months from an internet router communication network (Leskovec et al., 2005). We then use the top two eigenvectors of the normalized Laplacian matrix to give us node embeddings into $\mathbb{R}^2$ for each graph which gives us 9 datasets, one for each graph. Each dataset has roughly $n \sim 10^4$ points. This is an instance of spectral clustering.
**2) PHY**: Dataset from KDD cup 2004 (kdd, 2004). We take $10^4$ random samples to form our dataset.
**3) CIFAR10**: Testing portion of CIFAR-10 ($n = 10^4$, dimension 3072) (Krizhevsky, 2009).

**Baselines.**    We compare against the following algorithms. Additional experimental results on Lloyd's heuristic are given in Section E.3 in the supplementary material.

**1) kmeans++**: We measure the performance of our algorithm in comparison to the kmeans++ seeding algorithm. Since kmeans++ is a randomized algorithm, we take the average clustering cost after running kmeans++ seeding on 20 independent trials. We then standardize this value to have cost 1.0 and report all other costs in terms of this normalization. For example, the cost 2.0 means that the clustering cost is twice that of the average kmeans++ clustering cost. We also use the labels of kmeans++ *as the predictor* in the input for Algorithm 1 (denoted as "Alg + kmeans++") which serves to highlight the fact that one can use any heuristic or approximate clustering algorithm as a predictor.

**2) Random sampling**: For this algorithm, we subsample the predictor labels with probability $q$ ranging from 1% to 50%. We then construct the $k$-means centers using the labels of the sampled points and measure the clustering cost using the whole dataset. We use the *best* value of $q$ in our range every time to give this baseline as much power as possible. We emphasize that random sampling cannot have theoretical guarantees since the random sample can be corrupted (similarly

as in the example in Section 1.1). Thus some outlier detection steps (such as our algorithms) are required.

**Predictor Description.** We use the following predictors in our experiments.

**1) Nearest neighbor**: We use this predictor for the Oregon dataset. We find the best clustering of the node embeddings in Graph #1. In practice, this means running many steps of Lloyd's algorithm until convergence after initial seeding by `kmeans++`. Our predictor takes as input a point in $\mathbb{R}^2$ representing a node embedding of any of the later 8 graphs and outputs the label of the closest node in Graph #1.

**2) Noisy predictor.** This is the main predictor for PHY. We form this predictor by first finding the best $k$-means solution on our datasets. This again means initial seeding by `kmeans++` and then many steps of Lloyd's algorithm until convergence. We then randomly corrupt the resulting labels by changing them to a uniformly random label independently with error probability ranging from 0 to 1. We report the cost of clustering using only these noisy labels versus processing these labels using Algorithm 1.

**3) Neural network.** We use a standard neural network architecture (ResNet18) trained on the training portion of the CIFAR-10 dataset as the oracle for the testing portion which we use in our experiments. We used a pretrained model obtained from Huy (2020). Note that the neural network is predicting the class of the input image. However, the class value is highly correlated with the optimal $k$-means cluster group.

**Summary of results.** Our experiments show that our algorithm can leverage predictors to significantly improve the cost of $k$-means clustering and that good predictors can be easily tailored to the data at hand. The cost of $k$-means clustering reduces significantly after applying our algorithm compared to just using the predictor labels for two of our predictors. Lastly, the quality of the predictor remains high for the Oregon dataset even though the later graphs have changed and "moved away" from Graph #1.

**Selecting $\alpha$ in Algorithm 2.** In practice, the choice of $\alpha$ to use in our algorithm depends on the given predictor whose properties may be unknown. Since our goal is to minimize the $k$-means clustering objective (1), we can simply pick the 'best' value. To do so, we iterate over a small range of possible $\alpha$ from .01 to .15 in Algorithm 2 with a step size of 0.01 and select the clustering that results in the lowest objective cost. The range is fixed for all of our experiments. (See Paragraph 2.1

## 4.1 RESULTS

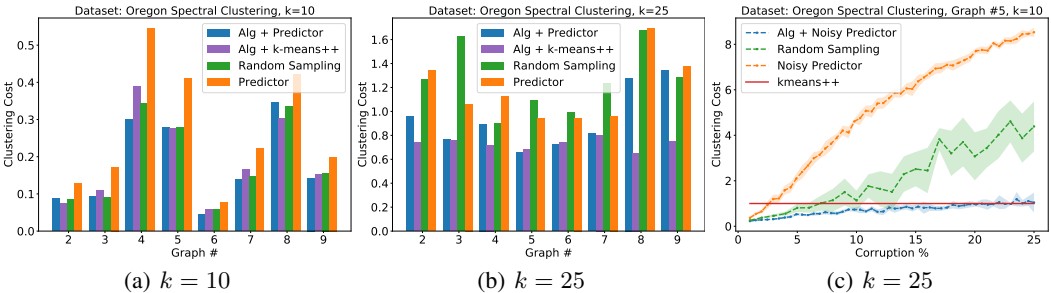

(a) $k = 10$      (b) $k = 25$      (c) $k = 25$

Figure 1: Performance of Algorithm 1 on later graph embeddings using Graph #1 as predictor.

**Oregon.** We first compare our algorithm with Graph #1 as the predictor against various baselines. This is shown in Figures 1(a) and Figure 1(b). In the $k = 10$ case, Figure 1(a) shows that the predictor returns a clustering better than using just the `kmeans++` seeding, which is normalized to have cost 1.0. This is to be expected since the subsequent graphs represent a similar network as Graph #1, just sampled later in time. However, the clustering improves significantly after using our algorithm on the predictor labels as the average cost drops by **55**%. We also see that using our algorithm after `kmeans++` is also sufficient to give significant decrease in clustering cost. Lastly,

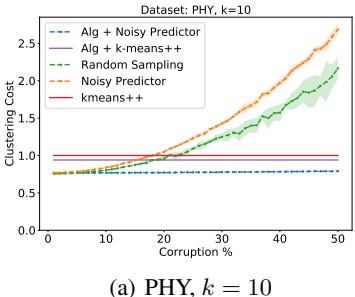 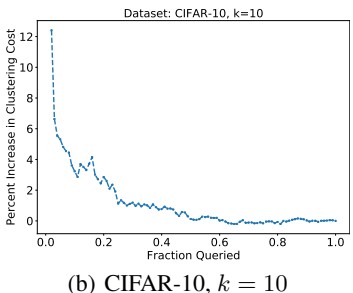

(a) PHY, $k = 10$          (b) CIFAR-10, $k = 10$

Figure 2: Our algorithm is able to recover a good clustering even for very high levels of noise.

random sampling also gives comparable results. This can be explained because we are iterating over a large range of subsampling probabilities for random sampling.

In the $k = 25$ case, Figure 1(b) shows that the oracle performance degrades and is worse than the baseline in 5 of the 8 graphs. However our algorithm again improves the quality of the clustering over the oracle across all graphs. Using kmeans++ as the predictor in our algorithm also improves the cost of clustering. The performance of random sampling is also worse. For example in Graph #3 for $k = 25$, it performed the worst out of all the tested algorithms.

Our algorithm also remains competitive with kmeans++ seeding even if the predictor for the Oregon dataset is highly corrupted. We consider a later graph, Graph #5, and corrupt the labels of the predictor randomly with probability $q$ ranging from 1% to 25% for the $k = 10$ case in Figure 1(c). While the cost of clustering using just the predictor labels can become increasingly worse, our algorithm is able to sufficiently "clean" the predictions. In addition, the cost of random sampling also gets worse as the corruptions increase, implying that it is much more sensitive to noise than our algorithm. The qualitatively similar plot for $k = 25$ is given in the supplementary section. Note that in spectral clustering, one may wish to get a mapping to $\mathbb{R}^d$ for $d > 2$. We envision that our results translate to those settings as well since having higher order spectral information only results in a stronger predictor. We continue the discussion on the PHY and CIFAR-10 datasets in Section E.

**Comparison to Lloyd's Heuristic.** In Section E.3, we provide additional results on experiments using Lloyd's heuristic. In summary, we give both theoretical and empirical justifications for why our algorithms are superior to blindly following a predictor and then running Lloyd's heuristic.

## ACKNOWLEDGEMENTS

Zhili Feng, David P. Woodruf, and Samson Zhou would like to thank partial support from NSF grant No. CCF- 181584, Office of Naval Research (ONR) grant N00014-18-1-256, and a Simons Investigator Award. Sandeep Silwal was supported in part by a NSF Graduate Research Fellowship Program.

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

# A    APPENDIX

**Theorem A.1** (Chernoff Bounds). *Let $X_1, \ldots, X_n$ be independent random variables taking values in $\{0, 1\}$. Let $X = \sum_{i=1}^{n} X_i$ denote their sum and let $\mu = \mathbb{E}[X]$ denote the sum's expected value. Then for any $\delta \in (0, 1)$ and $t > 0$,*

$$\mathbf{Pr}\left[X \leq (1 - \delta)\mu\right] \leq e^{-\frac{\delta^2 \mu}{2}}.$$

*For any $\delta > 0$,*

$$\mathbf{Pr}\left[X \geq (1 + \delta)\mu\right] \leq e^{-\frac{\delta^2 \mu}{3}}.$$

*Furthermore,*

$$\mathbf{Pr}\left[|X - \mu| \geq t\right] \leq e^{-\frac{t^2}{4n}}.$$

## A.1    PROOF OF THEOREM 2.1

We first prove Theorem 2.1, which shows that Algorithm 1 provides a $(1 + \alpha)$-approximation to the optimal $k$-means clustering, but uses suboptimal time compared to a faster algorithm we present in Section 3. All omitted proofs of lemmas appear in Section A.2.

We first show that for each coordinate, the empirical center for any $(1 - \alpha)$-fraction of the input points provides a good approximation to the optimal $k$-means clustering cost.

**Lemma A.2.** *Let $P, Q \subseteq \mathbb{R}$ be sets of points on the real line such that $|P| \geq (1 - \alpha)n$ and $|Q| \leq \alpha n$. Let $X = P \cup Q$, $C_P$ be the mean of $P$ and $C_X$ be the mean of $X$. Then $\text{cost}(X, C_P) \leq \left(1 + \frac{\alpha}{1 - \alpha^2}\right) \text{cost}(X, C_X)$.*

We now show that a conceptual interval $I^* \subset \mathbb{R}$ with "small" length contains a significant fraction of the true points. Ultimately, we will show that the interval $I$ computed in the "training" phase in CRDEST has smaller length than $I^*$ with high probability and yet $I$ also contains a significant fraction of the true points. The main purpose of $I^*$ (and eventually $I$) is to filter out extreme outliers because the "testing" phase only considers points in $I \cap X_2$.

**Lemma A.3.** *For a fixed set $X \subseteq \mathbb{R}$, let $C$ be the mean of $X$ and $\sigma^2 = \frac{1}{2|X|} \sum_{x \in X} (x - C)^2$ be the variance. Then the interval $I^* = \left[C - \frac{\sigma}{\sqrt{\alpha}}, C + \frac{\sigma}{\sqrt{\alpha}}\right]$ contains at least a $(1 - 4\alpha)$ fraction of the points in $X$.*

Using Lemma A.3, we show that the interval $I$ that is computed in the "training" phase contains a significant fraction of the true points.

**Lemma A.4.** *Let $m$ be a sufficiently large consatnt. We have that $I := [a, b]$ contains at least a $1 - 6\alpha$ fraction of points of $X_2$ and $b - a \leq 2\sigma/\sqrt{\alpha}$, with high probability, i.e., $1 - 1/\text{poly}(m)$.*

We next show that the optimal clustering on a subset obtained by independently sampling each input point provides a rough approximation of the optimal clustering. That is, the optimal center is well-approximated by the empirical center of the sampled points.

**Lemma A.5.** *Let $S$ be a set of points obtained by independently sampling each point of $X \subseteq \mathbb{R}^d$ with probability $p = \frac{1}{2}$. Let $C$ be the optimal center of these points and $C_S$ be the empirical center of these points. Conditioned on $|S| \geq 1$, then $\mathbb{E}[C_S] = \overline{x}$ and there exists a constant $\gamma$ such that for $\eta \geq 1$ and $|X| > \frac{\eta \gamma k}{\alpha}$,*

$$\mathbb{E}\left[\|C_S - \overline{x}\|_2^2\right] \leq \frac{\gamma}{|X|^2} \cdot \left(\sum_{x \in X} \|x - \overline{x}\|_2^2\right)$$

$$\mathbf{Pr}\left[\text{cost}(X, C_S) > (1 + \alpha)\text{cost}(X, C)\right] < 1/(\eta k).$$

Using Lemma A.2, Lemma A.4, and Lemma A.5, we justify the correctness of the subroutine CRDEST.

**Lemma A.6.** *Let $\alpha \in (10 \log n/\sqrt{n}, 1/7)$. Let $P, Q \subseteq \mathbb{R}$ be sets of points on the real line such that $|P| \geq (1 - \alpha)2m$ and $|Q| \leq 2\alpha m$, and $X = P \cup Q$. Let $C$ be the center of $P$. Then CRDEST on input set $X$ outputs a point $C'$ such that with probability at least $1 - 1/(\eta k)$, $\mathrm{cost}(P, C') \leq (1 + 18\alpha)(1 + \alpha)\left(1 + \frac{\alpha}{(1-\alpha)^2}\right)\mathrm{cost}(P, C)$.*

Using CRDEST as a subroutine for each coordinate, we now prove Theorem 2.1, justifying the correctness of Algorithm 1 by generalizing to all coordinates and centers and analyzing the runtime of Algorithm 1.

*Proof of Theorem 2.1.* Since $\Pi$ has label error rate $\lambda \leq \alpha$, then by definition of label error rate, at least a $(1 - \alpha)$ fraction of the points in each cluster are correctly labeled. Note that the $k$-means clustering cost can be decomposed into the sum of the costs induced by the centers in each dimension. Specifically, for a set $\mathcal{C} = \{C_1, \ldots, C_k\}$ of optimal centers,

$$\mathrm{cost}(X, \mathcal{C}) := \sum_{x \in X} d(x, \mathcal{C})^2 = \sum_{i=1}^{k} \sum_{x \in S_i} d(x, C_i)^2,$$

where $S_i$ is the set of points in $X$ that are assigned to center $C_i$. For a particular $i \in [k]$, we have

$$\sum_{x \in S_i} d(x, C_i)^2 = \sum_{x \in S_i} \sum_{j=1}^{d} d(x_j, (C_i)_j)^2,$$

where $x_j$ and $(C_i)_j$ are the $j$-th coordinate of $x$ and $C_i$, respectively.

By Lemma A.6, the cost induced by CRDEST for each dimension in each center $C_i'$ is a $(1 + \alpha)$-approximation of the total clustering cost for the optimal center $C_i$ in that dimension with probability $1 - 1/(\eta k)$. That is,

$$\sum_{x \in S_i} d(x_j, (C_i')_j)^2 \leq (1 + 18\alpha)(1 + \alpha)(1 + \alpha/(1 - \alpha)^2) \sum_{x \in S_i} d(x_j, (C_i)_j)^2$$

for each $j \in [d]$. Thus, taking a sum over all dimensions $j \in [d]$ and union bounding over all centers $i \in [k]$, we have that the total cost induced by Algorithm 1 is a $(1 + 20\alpha)$-approximation to the optimal $k$-means clustering cost with probability at least $1 - 1/\eta$.

To analyze the time complexity of Algorithm 1, first consider the subroutine CRDEST. It takes $O(kdn)$ time to first split each of the points in each cluster and dimension into two disjoint groups. Finding the smallest interval that contains a certain number of points can be done by first sorting the points and then iterating from the smallest point to the largest point and taking the smallest interval that contains enough points. This requires $O(n \log n)$ time for each dimension and each center, which results in $O(kdn \log n)$ total time. Once each of the intervals is found, computing the approximate center then takes $O(kdn)$ total time. Hence, the total running time of Algorithm 1 is $O(kdn \log n)$. □

## A.2 PROOF OF AUXILIARY LEMMAS

**Lemma A.2.** *Let $P, Q \subseteq \mathbb{R}$ be sets of points on the real line such that $|P| \geq (1 - \alpha)n$ and $|Q| \leq \alpha n$. Let $X = P \cup Q$, $C_P$ be the mean of $P$ and $C_X$ be the mean of $X$. Then $\mathrm{cost}(X, C_P) \leq \left(1 + \frac{\alpha}{1 - \alpha^2}\right)\mathrm{cost}(X, C_X)$.*

*Proof.* Suppose without loss of generality, that $C_X = 0$ and $C_P \leq 0$, so that $C_Q \geq 0$, where $C_Q$ is the mean of $Q$. Then it is well-known, e.g., see Inaba et al. (1994), that

$$\mathrm{cost}(X, C_P) = \mathrm{cost}(X, C_X) + |X| \cdot |C_P - C_X|^2.$$

Hence, it suffices to show that $|X| \cdot |C_P - C_X|^2 \leq \frac{\alpha}{(1-\alpha)^2}\mathrm{cost}(X, C_X)$.

Since $C_X = 0$ we have $|P| \cdot C_P = -|Q| \cdot C_Q$, with $|P| \geq (1-\alpha)n$ and $|Q| \leq \alpha n$. Let $|P| = (1-\varrho)n$ and $|Q| = \varrho n$ for some $\varrho \leq \alpha$. Thus, $C_Q = -\frac{1-\varrho}{\varrho} \cdot C_P$. By convexity, we thus have that

$$
\begin{aligned}
\mathrm{cost}(Q, C_X) &\geq |Q| \cdot \frac{(1-\varrho)^2}{\varrho^2} \cdot |C_P|^2 \\
&= \frac{n(1-\varrho)^2}{\varrho} \cdot |C_P|^2 \\
&\geq \frac{n(1-\alpha)^2}{\alpha} \cdot |C_P|^2.
\end{aligned}
$$

Therefore, we have

$$
|C_P - C_X|^2 = |C_P|^2 \leq \frac{\alpha}{n(1-\alpha)^2} \mathrm{cost}(Q, C_X) \leq \frac{\alpha}{n(1-\alpha)^2} \mathrm{cost}(X, C_X).
$$

Thus,

$$
|X| \cdot |C_P - C_X|^2 \leq \frac{\alpha}{(1-\alpha)^2} \mathrm{cost}(X, C_X),
$$

as desired. $\square$

**Lemma A.3.** *For a fixed set $X \subseteq \mathbb{R}$, let $C$ be the mean of $X$ and $\sigma^2 = \frac{1}{2|X|} \sum_{x \in X} (x - C)^2$ be the variance. Then the interval $I^* = \left[ C - \frac{\sigma}{\sqrt{\alpha}}, C + \frac{\sigma}{\sqrt{\alpha}} \right]$ contains at least a $(1 - 4\alpha)$ fraction of the points in $X$.*

*Proof.* Note that any point $x \in X \setminus I^*$ satisfies $|x - C|^2 > \sigma^2/(4\alpha)$. Thus, if more than a $4\alpha$ fraction of the points of $X$ are outside of $I^*$, then the total variance is larger than $\sigma^2$, which is a contradiction. $\square$

For ease of presentation, we analyze $\lambda = \frac{1}{2}$ and we note that the analysis extends easily to general $\lambda$. We now prove the technical lemma that we will use in the proof of Lemma A.8.

**Lemma A.7.** *We have*

$$
\sum_{j=1}^{m} \frac{\binom{m}{j}}{j \cdot 2^m} = \Theta\left(\frac{1}{m}\right).
$$

*Proof.* Let $m$ be sufficiently large. A Chernoff bound implies that for a sufficiently large constant $C$,

$$
\sum_{|j - m/2| \geq C\sqrt{m}} \frac{\binom{m}{j}}{2^m} \leq \frac{1}{m^2}.
$$

Furthermore,

$$
\sum_{j \geq C'm} \frac{\binom{m}{j}}{j \cdot 2^m} = O\left(\frac{1}{m}\right) \cdot \sum_{j \geq 1} \frac{\binom{m}{j}}{2^m} = O\left(\frac{1}{m}\right)
$$

so the upper bound on the desired relation holds. A similar analysis provides a lower bound. $\square$

**Lemma A.4.** *Let $m$ be a sufficiently large consatnt. We have that $I := [a, b]$ contains at least a $1 - 6\alpha$ fraction of points of $X_2$ and $b - a \leq 2\sigma/\sqrt{\alpha}$, with high probability, i.e., $1 - 1/\mathrm{poly}(m)$.*

*Proof.* By Lemma A.3, $I^*$ contains at least $2m(1 - 4\alpha)$ of the points in $X$. Hence, by applying an additive Chernoff bound for $t = O(\sqrt{m \log m})$ and for sufficiently large $m$, we have that the number of points in $I^* \cap X_1$ is at least $m(1 - 5\alpha)$ with high probability. Since $I$ is the interval of *minimal* length with at least $m(1 - 5\alpha)$ points, then the length of $I$ is at most the length of $I^*$. Moreover, again applying Chernoff bounds, we have that the number of points in $I \cap X_2$ is at least $m(1 - 6\alpha)$.

More formally, suppose we have a set of $2m$ points that we randomly partition into two sets $X_1$ and $X_2$. Consider any fixed interval $J$ that has at least $2cm$ total points for $c \geq 1 - 5\alpha$ (note there

are at most $O(m^2)$ intervals in total since our points are in one dimension). Let $J_1$ and $J_2$ denote the number of points in $J$ that are in $X_1$ and $X_2$ respectively. By a Chernoff bound, we have that both $J_1$ and $J_2$ are at least $mc(1-\alpha)$ with high probability. In particular, $|J_1 - J_2| \le \alpha mc$ with high probability. Thus by using a union bound, all intervals with at least $cm$ total points satisfy the property that the number of points partitioned to $X_1$ and the number of points partitioned to $X_2$ differ by at most $\alpha mc$ with high probability. Conditioning on this event, $I$ must also contain $m(1-6\alpha)$ points in $X_2$ since it contains at least $m(1-5\alpha)$ points in $X_1$, as desired. $\qquad\square$

**Lemma A.8.** *Let $S$ be a set of points obtained by independently sampling each point of $X \subseteq \mathbb{R}^d$ with probability $\frac{1}{2}$, and let $C_S$ be the centroid of $S$. Let $\overline{x}$ be the centroid of $X$. Conditioned on $|S| \ge 1$, we have $\mathbb{E}[C_S] = \overline{x}$, and there exists a constant $\gamma$ such that*

$$\mathbb{E}\left[\|C_S - \overline{x}\|_2^2\right] \le \frac{\gamma}{|X|^2} \cdot \left(\sum_{x \in X} \|x - \overline{x}\|_2^2\right).$$

*Proof.* We first prove that $\mathbb{E}[C_S] = \overline{x}$. Note that by the law of iterated expectations,

$$\mathbb{E}[C_S] = \mathbb{E}_{|S|}\mathbb{E}[C_S \mid |S|].$$

Let $x_{i_1}, \dots, x_{i_{|S|}}$ be a random permutation of the elements in $S$, so that for each $1 \le j \le |S|$, we have $\mathbb{E}[x_{i_j}] = \overline{x}$. Now conditioning on the size of $S$, we can write

$$C_S = \frac{x_{i_1} + \dots + x_{i_{|S|}}}{|S|}.$$

Therefore,

$$\mathbb{E}[C_S \mid |S|] = \frac{\overline{x} \cdot |S|}{|S|} = \overline{x}$$

and it follows that $\mathbb{E}[C_S] = \overline{x}$.

To prove that

$$\mathbb{E}\left[\|C_S - \overline{x}\|^2\right] \le \frac{\gamma}{|X|^2} \cdot \left(\sum_{x \in X} \|x - \overline{x}\|^2\right),$$

we again condition on $|S|$. Suppose that $|S| = j$. Then,

$$C_S - \overline{x} = \frac{(x_{i_1} - \overline{x}) + \dots + (x_{i_j} - \overline{x})}{j}$$

Now let $y_{i_t} = x_{i_t} - \overline{x}$ for all $1 \le t \le j$. Therefore,

$$\mathbb{E}_{|S|=j}\left[\|C_S - \overline{x}\|^2\right] = \frac{1}{j^2} \cdot \mathbb{E}\left[\|y_{i_1} + \dots + y_{i_j}\|^2\right]$$

$$= \frac{1}{j} \cdot \mathbb{E}[\|y_{i_1}\|^2] + \frac{j-1}{j} \cdot \mathbb{E}[y_{i_1}^T y_{i_2}].$$

Note that $x_{i_1}$ is uniform over elements in $X$, so it follows that

$$\mathbb{E}[\|y_{i_1}\|^2] = \frac{1}{|X|} \sum_{x \in X} \|x - \overline{x}\|^2.$$

Now if $j \ge 2$, we have that

$$\mathbb{E}[y_{i_1}^T y_{i_2}] = \frac{\sum_{a<b} y_a^T y_b}{\binom{|X|}{2}} = \frac{\|\sum_i y_i\|^2 - \sum_i \|y_i\|^2}{|X|(|X|-1)} \le 0$$

since $\sum_i y_i = 0$ by definition. Hence,

$$\mathbb{E}_{|S|\ge 2}\left[\|C_S - \overline{x}\|^2\right] \le \frac{1}{j \cdot |X|} \sum_{x \in X} \|x - \overline{x}\|^2.$$

Now the probability that $|S| = j$ for $j \geq 2$ is precisely $\binom{|X|}{j}/2^{|X|}$, so we have

$$\mathbf{Pr}\left[|S| \geq 2\right] \cdot \underset{|S| \geq 2}{\mathbb{E}}\left[\|C_S - \overline{x}\|^2\right]$$

$$\leq \frac{1}{|X|} \cdot \left(\sum_{x \in X} \|x - \overline{x}\|^2\right) \cdot \sum_{j=1}^{|X|} \frac{\binom{|X|}{j}}{j \cdot 2^{|X|}}.$$

From Lemma A.7, we have that

$$\sum_{j=1}^{|X|} \frac{\binom{|X|}{j}}{j \cdot 2^{|X|}} \leq \frac{c}{|X|}$$

for some constant $c$ so it follows that

$$\mathbb{E}\|C_S - \overline{x}\|^2 \leq \frac{c'}{|X|^2} \cdot \left(\sum_{x \in X} \|x - \overline{x}\|^2\right)$$

for some constant $c'$.

For $j = 1$, note that

$$\underset{|S|=j=1}{\mathbb{E}}\left[\|C_S - \overline{x}\|^2\right] = \frac{1}{|X|} \sum_{x \in X} \|x - \overline{x}\|^2.$$

Moreover, we have $\mathbf{Pr}\left[|S| = 1\right] = \frac{|X|}{2^{|X|}}$ and $\mathbf{Pr}\left[|S| = 0\right] = \frac{1}{2^{|X|}}$. Thus from the law of total expectation, we have

$$\mathbb{E}\left[\|C_S - \overline{x}\|^2\right] = \mathbf{Pr}\left[|S| < 2\right] \cdot \underset{|S|<2}{\mathbb{E}}\left[\|C_S - \overline{x}\|^2\right]$$

$$+ \mathbf{Pr}\left[|S| \geq 2\right] \cdot \underset{|S| \geq 2}{\mathbb{E}}\left[\|C_S - \overline{x}\|^2\right]$$

$$\leq \frac{|X|}{2^{|X|}} \cdot \frac{1}{|X|} \sum_{x \in X}\left(\|x - \overline{x}\|^2\right)$$

$$+ \frac{c'}{|X|^2} \cdot \left(\sum_{x \in X} \|x - \overline{x}\|^2\right)$$

$$\leq \frac{\gamma}{|X|^2} \cdot \left(\sum_{x \in X} \|x - \overline{x}\|^2\right)$$

for some constant $\gamma$, as desired. $\qquad\square$

**Lemma A.9.** *Let $S$ be a set of points obtained by independently sampling each point of $X \subseteq \mathbb{R}^d$ with probability $p = \frac{1}{2}$. Let $C$ be the optimal center of $X$ and $C_S$ be the empirical center of $S$. Let $\gamma \geq 1$ be the constant from Lemma A.8. Then for $\eta \geq 1$ and $|X| > \frac{\eta \gamma k}{\alpha}$,*

$$\mathbf{Pr}\left[\text{cost}(X, C_S) > (1 + \alpha) \text{cost}(X, C)\right] < 1/(\eta k).$$

*Proof.* By Lemma A.8 and Markov's inequality, we have

$$\mathbf{Pr}\left[\|C_S - C\|_2^2 \geq \frac{\eta \gamma k}{|X|^2} \sum_{x \in X} x^2\right] \leq \frac{1}{\eta k}.$$

We have

$$\sum_{x \in X} \|x - C_S\|_2^2 = \sum_{x \in X} \|x - C\|_2^2 + |X| \cdot \|C - C_S\|_2^2,$$

so that by Lemma A.8

$$\sum_{x \in X} \|x - C_S\|_2^2 \leq \left(1 + \frac{\eta \gamma k}{|X|}\right) \sum_{x \in X} \|x - C\|_2^2$$

$$= \left(1 + \frac{\eta \gamma k}{|X|}\right) \text{cost}(X, C),$$

with probability at least $1 - \frac{1}{\eta k}$. Hence for $|X| \geq \frac{\eta \gamma k}{\alpha}$, the approximate centroid of each cluster induces a $(1 + \alpha)$-approximation to the cost of the corresponding cluster. $\qquad \square$

**Lemma A.5.** *Let $S$ be a set of points obtained by independently sampling each point of $X \subseteq \mathbb{R}^d$ with probability $p = \frac{1}{2}$. Let $C$ be the optimal center of these points and $C_S$ be the empirical center of these points. Conditioned on $|S| \geq 1$, then $\mathbb{E}[C_S] = \overline{x}$ and there exists a constant $\gamma$ such that for $\eta \geq 1$ and $|X| > \frac{\eta \gamma k}{\alpha}$,*

$$\mathbb{E}\left[\|C_S - \overline{x}\|_2^2\right] \leq \frac{\gamma}{|X|^2} \cdot \left(\sum_{x \in X} \|x - \overline{x}\|_2^2\right)$$

$$\mathbf{Pr}\left[\text{cost}(X, C_S) > (1 + \alpha)\text{cost}(X, C)\right] < 1/(\eta k).$$

*Proof.* Lemma A.5 follows immediately from Lemma A.8 and Lemma A.9. $\qquad \square$

**Lemma A.6.** *Let $\alpha \in (10 \log n/\sqrt{n}, 1/7)$. Let $P, Q \subseteq \mathbb{R}$ be sets of points on the real line such that $|P| \geq (1 - \alpha)2m$ and $|Q| \leq 2\alpha m$, and $X = P \cup Q$. Let $C$ be the center of $P$. Then CRDEST on input set $X$ outputs a point $C'$ such that with probability at least $1 - 1/(\eta k)$, $\text{cost}(P, C') \leq (1 + 18\alpha)(1 + \alpha)\left(1 + \frac{\alpha}{(1-\alpha)^2}\right)\text{cost}(P, C)$.*

*Proof.* Let $\alpha \in (10 \log n/\sqrt{n}, 1/7)$. Then from Lemma A.4, we have that $I \cap X$ contains at least $(1 - 6\alpha)m$ points of $P \cap X_2$ and at most $2\alpha m$ points of $Q$ in an interval of length $2\sigma/\sqrt{\alpha}$, where

$$\sigma^2 = \frac{1}{2|P|} \sum_{p \in p} (p - C)^2 = \frac{1}{2|P|} \cdot \text{cost}(P, C).$$

From Lemma A.2, we have that

$$\text{cost}(P, C_0) \leq \left(1 + \frac{\alpha}{(1 - \alpha)^2}\right)\text{cost}(P, C_1),$$

where $C_0$ is the center of $I \cap P \cap X_2$ and $C_1$ is the center of $P \cap X_2$.

For sufficiently large $m$ and from Lemma A.9, we have that

$$\text{cost}(P, C_1) \leq (1 + \alpha)\text{cost}(P, C),$$

with probability at least $1 - 1/(\eta k)$. Thus, it remains to show that $\text{cost}(P, C') \leq (1 + O(\alpha))\text{cost}(P, C_0)$.

Since $C_0$ is the center of $I \cap P \cap X_2$ and $C'$ is the center of $I \cap X_2$, then we have

$$|I \cap P \cap X_2|C_0 + \sum_{q \in I \cap Q \cap X_2} q = |I \cap X_2|C'.$$

Since $I$ has length $2\sigma/\sqrt{\alpha}$, then $q \in \left[C_0 - \frac{2\sigma}{\sqrt{\alpha}}, C_0 + \frac{2\sigma}{\sqrt{\alpha}}\right]$. Because $|I \cap P \cap X_2| \geq (1 - 6\alpha)m$ and $|Q| = 2\alpha m$, then for sufficiently small $\alpha$, we have that

$$|C' - C_0| \leq 6\sqrt{\alpha}\sigma.$$

Note that we have $\text{cost}(P, C') = \text{cost}(P, C_0) + |P| \cdot |C_0 - C'|^2$, so that

$$\text{cost}(P, C') \leq \text{cost}(P, C_0) + |P| \cdot 36\alpha\sigma^2.$$

Finally, $\sigma^2 = \frac{1}{2|P|} \cdot \text{cost}(P, C)$ and $\text{cost}(P, C) \leq \text{cost}(P, C_0)$ due to the optimality of $C$. This implies

$$\text{cost}(P, C') \leq \text{cost}(P, C_0) + |P| \cdot 36\alpha\sigma^2$$

$$\leq \text{cost}(P, C_0) + |P| \cdot 36\alpha \cdot \frac{1}{2|P|} \cdot \text{cost}(P, C)$$

$$\leq \text{cost}(P, C_0) + 18\alpha \text{cost}(P, C_0)$$

$$= (1 + 18\alpha)\text{cost}(P, C_0),$$

as desired. Thus putting things together, we have

$$\text{cost}(P, C') \leq (1 + 18\alpha)(1 + \alpha)\left(1 + \frac{\alpha}{(1-\alpha)^2}\right)\text{cost}(P, C).$$

$\qquad \square$

## A.3 PROOF OF THEOREM 3.4

We now give the proofs for optimal query complexity and runtime. We first require the following analogue to Lemma A.5:

**Lemma A.10.** *Let $S$ be a set of points obtained by independently sampling each point of $X \subseteq \mathbb{R}^d$ with probability $p = \min\left(1, \frac{100 \log k}{\alpha |S|}\right)$. Let $C$ be the optimal center of these points and $C_S$ be the empirical center of these points. Conditioned on $|S| \geq 1$, then $\mathbb{E}[C_S] = \overline{x}$ and for $|X| > \frac{\gamma k}{\alpha}$,*

$$\mathbb{E}\left[\|C_S - \overline{x}\|_2^2\right] \leq \frac{\gamma}{p|X|^2} \cdot \left(\sum_{x \in X} \|x - \overline{x}\|_2^2\right)$$

*for some constant $\gamma$.*

**Lemma A.11.** *For $\alpha \in (10 \log n / \sqrt{n}, 1/7)$, let $\Pi$ be a predictor with error rate $\lambda \leq \alpha/2$. If each cluster has at least $\gamma k \log k / \alpha$ points, then Algorithm 3 outputs a $(1 + 20\alpha)$-approximation to the $k$-means objective value with probability at least $3/4$.*

*Proof.* Since $S$ samples each of points independently with probability proportional to cluster sizes given by $\Pi$, for a fixed $i \in [k]$ at least $\frac{90 \log k}{\alpha}$ points with label $i$ are sampled, with probability at least $1 - \frac{1}{k^4}$ from Chernoff bounds. Let $\gamma_1, \ldots, \gamma_k$ be the empirical means corresponding to each of the sampled points with labels $1, \ldots, k$, respectively, and let $\Gamma_0 = \{\gamma_1, \ldots, \gamma_k\}$. Let $C_1, \ldots, C_k$ be centers of a $(1 + \alpha)$-approximate optimal solution $\mathcal{C}$ with corresponding clusters $X_1, \ldots, X_k$. By Lemma A.10, we have that

$$\mathbb{E}\left[\|C_i - \gamma_i\|_2^2\right] \leq \frac{\gamma}{p|X_i|^2} \cdot \left(\sum_{x \in X_i} \|x - C_i\|_2^2\right),$$

where $p = \min\left(1, \frac{100 \log k}{\alpha |S|}\right)$. By Markov's inequality, we have that

$$\sum_{i \in [k]} \|C_i - \gamma_i\|_2^2 \leq 100 \sum_{i \in [k]} \frac{\gamma}{p|X_i|^2} \cdot \left(\sum_{x \in X_i} \|x - C_i\|_2^2\right)$$

with probability at least $0.99$. Similar to the proof of Lemma A.9, we use the identity

$$\sum_{x \in X_i} \|x - \gamma_i\|_2^2 = \sum_{x \in X_i} \|x - C_i\|_2^2 + |X_i| \cdot \|C_i - \gamma_i\|_2^2.$$

Hence, we have that

$$\text{cost}(X, \Gamma_0) \leq (1 + \alpha) \cdot \text{cost}(X, C),$$

with probability at least $0.99$.

Suppose $\Pi$ has error rate $\lambda \leq \alpha$ and each error chooses a label uniformly at random from the $k$ possible labels. Then by definition of error rate, at most $\alpha/2$ fraction of the points are erroneously labeled for each cluster. Each cluster in the optimal $k$-means clustering of the predictor $\Pi$ has at least $n/(\zeta k)$ points, so that at least a $(1 - \alpha)$ fraction of the points in each cluster are correctly labeled. Thus, by the same argument as in the proof of Lemma A.6, we have that Algorithm 1 outputs a set of centers $C_1, \ldots, C_k$ such that for $\Gamma = \{C_1, \ldots, C_k\}$, we have

$$\text{cost}(X, \Gamma) \leq (1 + 18\alpha)\left(1 - \frac{\alpha}{(1 - \alpha)^2}\right) \cdot \text{cost}(X, \Gamma_0),$$

with sufficiently large probability.

Let $\mathcal{E}$ be the event that $\text{cost}(X, \Gamma) \leq (1 + \alpha)(1 + 18\alpha)\left(1 - \frac{\alpha}{(1 - \alpha)^2}\right) \cdot \text{cost}(X, \mathcal{C})$, so that $\mathbf{Pr}[\mathcal{E}] \geq 1 - 1/\text{poly}(k)$. Conditioned on $\mathcal{E}$, let $X_1$ be the subset of $X$ that is assigned the correct label by $\Pi$, and let $X_2$ be the subset of $X$ assigned the incorrect label. For each point $x \in X_1$ assigned the

correct label $\ell_x$ by $\Pi$, the closest center to $x$ in $\Gamma$ is $C_{\ell_x}$, so Algorithm 3 will always label $x$ with $\ell_x$. Thus,

$$\mathrm{cost}(X_1, \Gamma) \leq \mathrm{cost}(X, \Gamma) \leq (1+\alpha)(1+18\alpha)\left(1 - \frac{\alpha}{(1-\alpha)^2}\right) \cdot \mathrm{cost}(X, \mathcal{C}),$$

conditioned on $\mathcal{E}$. On the other hand, if $x \in X_2$ is assigned an incorrect label $\ell_x$ by $\Pi$, then the $(2, r)$-approximate nearest neighbor data assigns the label $p_x$ to $x$, where $\phi(C_{p_x})$ is the closest center to $\phi(x)$ in the projected space. Recall that $\phi$ is the composition map $\phi_1 \circ \phi_2$, where $\phi_1$ has a terminal dimension reduction with distortion $5/4$, and $\phi_2$ is a random JL linear map with distortion $5/4$. Thus the distance between $x$ and $C_{p_x}$ is a 2-approximation between $x$ and its closest center $C_i$. Hence, by assigning all points $x$ to their respective centers $C_{p_x}$, we have $d(x, C_{p_x}) \leq 2\,\mathrm{cost}(x, \Gamma)$. Since each point $x \in X$ is assigned the incorrect label with probability $\lambda \leq \alpha/2$, the expected cost of the labels assigned to $X_2$ is $\alpha\,\mathrm{cost}(X, \Gamma)$. By Markov's inequality, the cost of the labels assigned to $X_2$ is at most

$$10\alpha\,\mathrm{cost}(X, \Gamma) < 10\alpha(1+\alpha)\,\mathrm{cost}(X, \mathcal{C}),$$

with probability at least $1 - \frac{1}{5}$, conditioned on $\mathcal{E}$.

Therefore by a union bound, the total cost is at most $(1+20\alpha) \cdot \mathrm{cost}(X, \mathcal{C})$, with probability at least $3/4$. $\qquad\square$

We need the following theorems on the quality of the data structures utilized in Algorithm 3.

**Theorem A.12.** *Makarychev et al. (2019) For every set $C \subset \mathbb{R}^d$ of size $k$, a parameter $0 < \alpha < \frac{1}{2}$ and the standard Euclidean norm $d(\cdot, \cdot)$, there exists a terminal dimension reduction $f : C \to \mathbb{R}^{d'}$ with distortion $(1 + \alpha)$, where $d' = O\left(\frac{\log k}{\alpha^2}\right)$. The dimension reduction can be computed in polynomial time.*

**Theorem A.13.** *Indyk & Motwani (1998); Har-Peled et al. (2012); Andoni et al. (2018) For $\alpha > 0$, there exists a $(1+\alpha, r)$-ANN data structure over $\mathbb{R}$ equipped with the standard Euclidean norm that achieves query time $O\left(d \cdot \frac{\log n}{\alpha^2}\right)$ and space $S := O\left(\frac{1}{\alpha^2}\log\frac{1}{\alpha} + d(n+q)\right)$, where $q := \frac{\log n}{\alpha^2}$. The runtime of building the data structure is $O(S + ndq)$.*

We now prove Theorem 3.4.

**Theorem 3.4.** *Let $\alpha \in (10\log n/\sqrt{n}, 1/7)$, $\Pi$ be a predictor with label error rate $\lambda \leq \alpha$, and $\gamma \geq 1$ be a sufficiently large constant. If each cluster in the optimal $k$-means clustering of the predictor has at least $\gamma k \log k/\alpha$ points, then Algorithm 3 outputs a $(1 + 20\alpha)$-approximation to the $k$-means objective with probability at least $3/4$, using $O(nd\log n + \mathrm{poly}(k, \log n))$ total time.*

*Proof.* The approximation guarantee of the algorithm follows from Lemma A.11. To analyze the running time, we first note that we apply a JL matrix with dimension $O(\log n)$ to each of the $n$ points in $\mathbb{R}^d$, which uses $O(nd\log n)$ time. As a result of the JL embedding, each of the $n$ points has dimension $O(\log n)$. Thus, by Theorem A.12, constructing the terminal embedding uses $\mathrm{poly}(k, \log n)$ time. As a result of the terminal embedding, each of the $k$ possible centers has dimension $O(\log k)$. Hence, by Theorem A.13, constructing the $(2, r)$-ANN data structure for the $k$ possible centers uses $O(k\log^2 k)$ time. Subsequently, each query to the data structure uses $O(\log^2 k)$ time. Therefore, the overall runtime is $O(nd\log n + \mathrm{poly}(k, \log n))$. $\qquad\square$

### A.4 Remark on Truly-polynomial time algorithms vs. PTAS/PRAS.

**Remark A.14.** We emphasize that the runtime of our algorithm in Theorem 2.1 is truly polynomial in all input parameters $n, d, k$ and $1/\alpha$ (and even near-linear in the input size $nd$). Although there exist polynomial-time randomized approximation schemes for $k$-means clustering, e.g., Inaba et al. (1994); Feldman et al. (2007); Kumar et al. (2004), their runtimes all have exponential dependency on $k$ and $1/\alpha$, i.e., $2^{\mathrm{poly}(k, 1/\alpha)}$. However, this does not suffice for many applications, since $k$ and $1/\alpha$ should be treated as input parameters rather than constants. For example, it is undesirable to pay an exponential amount of time to linearly improve the accuracy $\alpha$ of the algorithm. Similarly, if the number of desired clusters $k = O(\log^2 n)$, then the runtime would be exponential. Thus we believe the exponential improvement of Theorem 2.1 over existing PRAS in terms of $k$ and $1/\alpha$ is significant.

## A.5 REMARK ON POSSIBLE INSTANTIATIONS OF PREDICTOR

**Remark A.15.** We can instantiate Theorem 2.1 with various versions of the predictor. Assume each cluster in the $(1 + \alpha)$-approximately optimal $k$-means clustering of the predictor has size at least $n/(\zeta k)$ for some tradeoff parameter $\zeta \in [1, (\sqrt{n})/(8k \log n)]$. Then the clustering quality and runtime guarantees of Theorem 2.1 hold if the predictor $\Pi$ is such that

1. $\Pi$ outputs the right label for each point independently with probability $1 - \lambda$ and otherwise outputs a random label for $\lambda \leq O(\alpha/\zeta)$,
2. $\Pi$ outputs the right label for each point independently with probability $1 - \lambda$ and otherwise outputs an *adversarial* label for $\lambda \leq O(\alpha/(k\zeta))$.

In addition, if the predictor $\Pi$ outputs *a failure symbol* when it fails, then for constant $\zeta > 0$, there exists an algorithm (see supplementary material) that outputs a $(1+\alpha)$-approximation to the $k$-means objective with probability at least $2/3$, even when $\Pi$ has failure rate $\lambda = 1 - 1/\operatorname{poly}(k)$. Note that this remark (but not Theorem 2.1) assumes that each of the $k$ clusters in the $(1 + \alpha)$-approximately optimal clustering has at least $\frac{n}{\zeta k}$ points. This is a natural assumption that the clusters are "roughly balanced" which often holds in practice, e.g., for Zipfian distributions.

# B DELETION PREDICTOR

In this section, we present a fast and simple algorithm for $k$-means clustering, given access to a label predictor $\Pi$ with deletion rate $\lambda$. That is, for each point, the predictor $\Pi$ either outputs a label for the point consistent with an optimal $k$-means clustering algorithm with probability $\lambda$, or outputs nothing at all (or a failure symbol $\bot$) with probability $1 - \lambda$. Since the deletion predictor fails explicitly, we can actually achieve a $(1 + \alpha)$-approximation even when $\lambda = 1 - \frac{1}{\operatorname{poly}(k)}$.

Our algorithm first queries all points in the input $X$. Although the predictor does not output the label for each point, for each cluster $C_i$ with a sufficiently large number of points, with high probability, the predictor assigns at least $\frac{\lambda}{2}|C_i|$ points of $C_i$ to the correct label. We show that if $|C_i| = \Omega\left(\frac{k}{\alpha}\right)$, then with high probability, the empirical center is a good estimator for the true center. That is, the $k$-means objective using the centroid of the points labeled $i$ is a $(1 + \alpha)$-approximation to the $k$-means objective using the true center of $C_i$. We give the full details in Algorithm 4.

To show that the empirical center is a good estimator for the true center, recall that a common approach for mean estimation is to sample roughly an $O\left(\frac{1}{\alpha^2}\right)$ number of points uniformly at random with replacement. The argument follows from observing that each sample is an unbiased estimator of the true mean, and repeating $O\left(\frac{1}{\alpha^2}\right)$ times sufficiently upper bounds the variance.

Observe that the predictor can be viewed as sampling the points from each cluster *without replacement*. Thus, for sufficiently large cluster sizes, we actually have a huge number of samples, which intuitively should sufficiently upper bound the variance. Moreover, the empirical mean is again an unbiased estimator of the true mean. Thus, although the above analysis does not quite hold due to dependencies between the number of samples and the resulting averaging term, we show that the above intuition does hold.

---

**Algorithm 4** Linear time $k$-means algorithm with access to a label predictor $\Pi$ with deletion rate $\lambda$.

---

**Input:** A point set $x \in X$ with labels given by a label predictor $\Pi$ with deletion rate $\lambda$.
**Output:** A $(1 + \alpha)$-approximate $k$-means clustering of $X$.
1: **for** each label $i \in [k]$ **do**
2:     Let $S_i$ be the set of points labeled $i$.
3:     $c_i \leftarrow \frac{1}{|S_i|} \cdot \sum_{x \in S_i} x$
4: **end for**
5: **for** all points $x \in X$ **do**
6:     **if** $x$ is unlabeled **then**
7:         $\ell_x \leftarrow \arg \min d(x, c_i)$
8:         Assign label $\ell_x$ to $x$.
9:     **end if**
10: **end for**

---

We first show that independently sampling points uniformly at random from a sufficiently large point set guarantees a $(1 + \alpha)$-approximation to the objective cost. Inaba et al. (1994); Ailon et al. (2018) proved a similar statement for sampling with replacement.

It remains to justify the correctness of Algorithm 4 by arguing that with high probability, the overall $k$-means cost is preserved up to a $(1+\alpha)$-factor by the empirical means. We also analyze the running time of Algorithm 4.

**Theorem B.1.** *If each cluster in the optimal $k$-means clustering of the predictor $\Pi$ has at least $\frac{3k}{\alpha}$ points, then Algorithm 4 outputs a $(1 + \alpha)$-approximation to the $k$-means objective with probability at least $\frac{2}{3}$, using $O(kdn)$ total time.*

*Proof.* We first justify the correctness of Algorithm 4. Suppose each cluster in the optimal $k$-means clustering of the predictor $\Pi$ has at least $\frac{3k}{\alpha}$ points. Let $\mathcal{C} = \{c_1, \ldots, c_k\}$ be the optimal centers selected by $\Pi$ and let $\mathcal{C}_S = \{c'_1, \ldots, c'_k\}$ be the empirical centers chosen by Algorithm 4. For each $i \in [k]$, let $C_i$ be the points of $X$ that are assigned to $C_i$ by the predictor $\Pi$. By Lemma A.9 with $\eta = 3$, the approximate centroid of a cluster induces a $(1 + \alpha)$-approximation to the cost of the corresponding cluster so that

$$\text{cost}(C_i, c'_i) \leq (1 + \alpha)\,\text{cost}(C_i, c_i),$$

with probability at least $1 - \frac{1}{3k}$. Taking a union bound over all $k$ clusters, we have that

$$\sum_{i \in [k]} \text{cost}(C_i, c'_i) \leq \sum_{i \in [k]} (1 + \alpha)\,\text{cost}(C_i, c_i),$$

with probability at least $\frac{2}{3}$. Equivalently, $\text{cost}(X, C) \leq (1 + \alpha)\,\text{cost}(X, C_S)$.

To analyze the running time of Algorithm 4, observe that the estimated centroids for all labels can be computed in $O(dn)$ time. Subsequently, assigning each unlabeled point to the closest estimated centroid uses $O(kd)$ time for each unlabeled point. Thus, the total running time is $O(kdn)$. $\qquad\square$

## C $k$-MEDIAN CLUSTERING

We first recall that a well-known result states that the geometric median that results from uniformly sampling a number of points from the input is a "good" approximation to the actual geometric median for the 1-median problem.

**Theorem C.1.** *Krauthgamer (2019) Given a set $P$ of $n$ points in $\mathbb{R}^d$, the geometric median of a sample of $O\left(\frac{d}{\alpha^2} \log \frac{d}{\alpha}\right)$ points of $P$ provides a $(1 + \alpha)$-approximation to the 1-median clustering problem with probability at least $1 - 1/\text{poly}(d)$.*

Note that we can first apply Theorem A.12 to project all points to a space with dimension $O\left(\frac{1}{\alpha^2} \log \frac{k}{\alpha}\right)$ before applying Theorem C.1. Instead of computing the geometric median, we recall the following procedure that produces a $(1 + \alpha)$-approximation to the geometric median.

**Theorem C.2.** *Cohen et al. (2016) There exists an algorithm that outputs a $(1 + \alpha)$-approximation to the geometric median in $O\left(nd \log^3 \frac{n}{\alpha}\right)$ time.*

We give our algorithm in full in Algorithm 5.

**Theorem C.3.** *For $\alpha \in (0, 1)$, let $\Pi$ be a predictor with error rate $\lambda = O\left(\frac{\alpha^4}{k \log \frac{k}{\alpha} \log \frac{\log k}{\alpha}}\right)$. If each cluster in the optimal $k$-median clustering of the predictor has at least $n/(\zeta k)$ points, then there exists an algorithm that outputs a $(1 + \alpha)$-approximation to the $k$-median objective with probability at least $1 - 1/\text{poly}(k)$, using $O(nd \log^3 n + \text{poly}(k, \log n))$ total time.*

*Proof.* Observe that Algorithm 5 samples $O\left(\frac{1}{\alpha^4} \log^2 \frac{k}{\alpha}\right)$ points for each of the clusters labeled $i$, with $i \in [k]$. Thus Algorithm 5 samples $O\left(\frac{k}{\alpha^4} \log^2 \frac{k}{\alpha}\right)$ points in total. For $\lambda = O\left(\frac{\alpha^4}{k \log \frac{k}{\alpha} \log \frac{\log k}{\alpha}}\right)$ with a sufficiently small constant, the expected number of incorrectly labeled points sampled by Algorithm 5 is less than $\frac{1}{32}$. Thus, by Markov's inequality, the probability that no incorrectly labeled

---

**Algorithm 5** Learning-Augmented $k$-median Clustering

---

**Input:** A point set $x \in X$ with labels given by a predictor $\Pi$ with error rate $\lambda$.
**Output:** A $(1 + \alpha)$-approximate $k$-median clustering of $X$.
1: Use a terminal embedding to project all points into a space with dimension $O\left(\frac{1}{\alpha^2}\log\frac{k}{\alpha}\right)$.
2: **for** $i = 1$ to $i = k$ **do**
3:     Let $\ell_i$ be the most common remaining label.
4:     Sample $O\left(\frac{1}{\alpha^4}\log^2\frac{k}{\alpha}\right)$ points with label $\ell_i$.
5:     Let $C_i'$ be a $\left(1 + \frac{\alpha}{4}\right)$-approximation to the geometric median of the sampled points.
6: **end for**
7: **Return** $C_1', \ldots, C_k'$.

---

points are sampled by Algorithm 5 is at least $\frac{3}{4}$. Conditioned on the event that no incorrectly labeled points are sampled by Algorithm 5, then by Theorem C.1, the empirical geometric median for each cluster induces a $\left(1 + \frac{\alpha}{4}\right)$-approximation to the optimal geometric median in the projected space. Hence the set of $k$ empirical geometric medians induces a $\left(1 + \frac{\alpha}{4}\right)$-approximation to the optimal $k$-median clustering cost in the projected space. Since the projected space is the result of a terminal embedding, the set of $k$ empirical geometric medians for the sampled points in the projected space induces a $k$-median clustering cost that is a $\left(1 + \frac{\alpha}{4}\right)$-approximation to the $k$-median clustering cost induced by the set of $k$ empirical geometric medians for the sampled points in the original space. Taking the set of $k$ empirical geometric medians for the sampled points in the original space induces a $\left(1 + \frac{\alpha}{4}\right)^2$-approximation to the $k$-median clustering cost. We take a $\left(1 + \frac{\alpha}{4}\right)$-approximation to each of the geometric medians. Thus for sufficiently small $\alpha$, Algorithm 5 outputs a $(1 + \alpha)$-approximation to the $k$-median clustering problem.

To embed the points into the space of dimension $O\left(\frac{1}{\alpha^2}\log\frac{k}{\alpha}\right)$, Algorithm 5 spends $O(nd\log n)$ total time. By Theorem C.2, it takes $O(nd\log^3 n)$ total time to compute the approximate geometric medians. $\qquad\square$

## D    LOWER BOUNDS

MAX-E3-LIN-2 is the optimization problem of maximizing the number of equations satisfied by a system of linear equations of $\mathbb{Z}_2$ with exactly 3 distinct variables in each equation. E$K$-MAX-E3-LIN-2 is the problem of MAX-E3-LIN-2 when each variable appears in exactly $k$ equations. Fotakis et al. (2016) showed that assuming the exponential time hypothesis (ETH) (Impagliazzo & Paturi, 2001), there exists an absolute constant $C_1$ such that MAX $k$-SAT (and thus MAX $k$-CSP) instances with fewer than $O(n^{k-1})$ clauses cannot be approximated within a factor of $C_1$ in time $2^{O(n^{1-\delta})}$ for any $\delta > 0$. As a consequence, the reduction by Håstad (2001) shows that there exist absolute constants $C_2, C_3$ such that E$K$-MAX-E3-LIN-2 with $k \geq C_2$ cannot be approximated within a factor of $C_3$ in time $2^{O(n^{1-\delta})}$ for any $\delta > 0$. Hence, the reduction by Chlebík & Chlebíková (2006) shows that there exists a constant $C_4$ such that approximating the minimum vertex cover of 4-regular graphs within a factor of $C_4$ cannot be done in time $2^{O(n^{1-\delta})}$ for any $\delta > 0$. Thus the reduction by Lee et al. (2017) shows that there exists a constant $C_5$ such that approximating $k$-means within a factor of $C_5$ cannot be done in time $2^{O(n^{1-\delta})}$ for any $\delta > 0$, assuming ETH. Namely, the reduction of Lee et al. (2017) shows that an algorithm that provides a $C_5$-approximation to the optimal $k$-means clustering can be used to compute a $C_4$-approximation to the minimum vertex cover.

**Theorem D.1.** *If ETH is true, then there does not exist an algorithm $\mathcal{A}$ that takes a set $S$ of $\frac{n^{1-\delta}}{\log n}$ vertices and finds a $C_4$-approximation to the minimum vertex cover that contains $S$ on a 4-regular graph $G$, using $2^{O(n^{1-\delta})}$ time for some constant $\delta \in (0, 1]$.*

*Proof.* Suppose by way of contradiction that there exists an algorithm $\mathcal{A}$ that takes a set $S$ of $\frac{n^{1-\delta}}{\log n}$ vertices and finds a $C_4$-approximation to the minimum vertex cover that contains $S$ on a 4-regular graph $G$, using $2^{O(n^{1-\delta})}$ time for some constant $\delta \in (0, 1]$. We claim that we can use $\mathcal{A}$ to create an overall algorithm that violates ETH. Indeed, suppose we guess each subset of $\frac{n^{1-\delta}}{\log n}$ ver-

tices and which vertices of the subset are in the cover. There are $\binom{n}{n^{1-\delta}/\log n} \cdot 2^{n^{1-\delta}/\log n} \leq$ $(en^\delta \log n)^{n^{1-\delta}/\log n} \cdot 2^{n/\log n}$ such combinations of vertices. For each guess, we then run the purported algorithm $\mathcal{A}$ that uses $2^{O(n^{1-\delta})}$ time. Thus we can identify a $C_4$-approximation to the minimum vertex cover in time

$$(en^\delta \log n)^{n^{1-\delta}/\log n} \cdot 2^{n/\log n} \cdot 2^{O(n^{1-\delta})} = 2^{O(n^{1-\delta})} \cdot 2^{O(n^{1-\delta})} = 2^{O(n^{1-\delta})},$$

which would contradict ETH. □

Finally, we show the query complexity of Algorithm 3 is nearly optimal. Lee et al. (2017) constructed an instance of $k$-means that cannot be approximated within a factor of 1.0013 in polynomial time. The reduction of Lee et al. (2017) creates $4n$ points in $\mathbb{R}^{3n}$ that must be clustered by $O(n)$ centers and an algorithm that provides a $C_5$-approximation to the optimal $k$-means clustering can be used to compute a $C_4$-approximation to the minimum vertex cover. Thus, there exists a constant $C_5$ such that approximating $k$-means within a factor of $C_5$ cannot be done in time $2^{O(n^{1-\delta})}$ for any $\delta > 0$, assuming ETH.

**Theorem 3.5.** *For any $\delta \in (0,1]$, any algorithm that makes $O\left(\frac{k^{1-\delta}}{\alpha \log n}\right)$ queries to the predictor with label error rate $\alpha$ cannot output a $(1+C\alpha)$-approximation to the optimal $k$-means clustering cost in time $2^{O(n^{1-\delta})}$ time, assuming the Exponential Time Hypothesis.*

*Proof.* Let $\alpha$ be a fixed constant such that $\alpha < C_5$. Given an instance $\mathcal{I}$ of a $k$-means clustering constructed from the reduction of Lee et al. (2017), the optimal clustering cost is $\Omega(n)$ and $k_1 = \Omega(n)$. We embed this instance into a $k$-means clustering by adding an additional $\Omega(n)$ points arbitrarily far from $\mathcal{I}$, so that the additional points contribute $\Omega(n/\alpha)$ cost upon partitioning into $k_2 = \Omega(n)$ clusters. We set $k = k_1 + k_2$.

In summary, the resulting instance has $O(n)$ points and $k = \Omega(n)$. The optimal solution has cost $O(n/\alpha)$ cost so that $\mathcal{I}$ contributes an $\Omega(\alpha)$ fraction of the cost. By querying $O\left(\frac{k^{1-\delta}}{\alpha \log n}\right)$ points with sufficiently small constant, at most $O\left(\frac{k^{1-\delta}}{\alpha \log n}\right)$ of the cluster centers in $\mathcal{I}$ will be revealed by the construction of $\mathcal{I}$. Each center corresponds to a selected vertex in the corresponding vertex cover in the reduction from minimum vertex cover on 4-regular graphs. Hence, in the corresponding vertex cover instance, at most $O\left(\frac{k^{1-\delta}}{\alpha \log n}\right)$ vertices are revealed. Thus by Theorem D.1, any algorithm running in $2^{O(n^{1-\delta})}$ time cannot determine a $C_5$-approximation to the optimal $k$-means clustering cost on $\mathcal{I}$, as it would correspond to a $C_4$-approximation to the optimal vertex cover, assuming ETH. Since $\mathcal{I}$ induces an $\Omega(\alpha)$ fraction of the total clustering cost, it follows that any algorithm that makes $O\left(\frac{k^{1-\delta}}{\alpha \log n}\right)$ queries cannot output a $(1+C\alpha)$-approximation to the optimal $k$-means clustering cost in time $2^{O(n^{1-\delta})}$ time, assuming ETH. □

# E  ADDITIONAL EXPERIMENTAL RESULTS

## E.1  OMITTED DISCUSSION

In this section we continue the discussion from the experimental section of the main body. In Figure 3(a), we show the qualitatively similar version of Figure 2(c) of the main body for the case of $k = 25$. In Figure 3(b), we display the qualitatively similar version of Figure 3(a) of the main body for the $k = 25$ case of dataset PHY.

**PHY.** We use the noisy predictor for this dataset. We see in Figure 2(a) that as the corruption percentage rises, the clustering given by just the predictor labels can have increasingly large cost. Nevertheless, even if the clustering cost of the corrupted labels is rising, the cost decreases significantly after applying Algorithm 1 by roughly a factor of **3x**. Indeed, we see that our algorithm can beat the kmeans++ seeding baseline for $q$ as high as $50\%$. Just as in Figure 1(c), random sampling is sensitive to noise. Lastly, we also remain competitive with the purple line which uses the labels output by kmeans++ as the predictor in our algorithm (no corruptions added). The qualitatively similar plot for $k = 25$ is given in the supplementary material.

**CIFAR-10.** The cost of clustering on CIFAR-10 using only the predictor, the predictor with our algorithm, random sampling, and `kmeans++` as the predictor for our algorithm were $0.733$, $0.697$, $0.721$, and $0.640$, respectively, where $1.0$ represents the cost of `kmeans++`. The neural network was very accurate ($\sim 93\%$) in predicting the class of the input points which is highly correlated with the optimal $k$-means clusters. Nevertheless, our algorithm improved upon this highly precise predictor.

Note that using `kmeans++` as the predictor for our algorithm resulted in a clustering that was **13%** **better** than the one given by the neural network predictor. This highlights the fact that an approximate clustering combined with our algorithm can be competitive against a highly precise predictor, such as a neural network, even though creating the highly accurate predictor can be expensive. Indeed, obtaining a neural network predictor requires prior training data and also the time to train the network. On the other hand, using a heuristic clustering as a predictor is extremely flexible and can be applied to any dataset even if no prior training dataset is available ($50,000$ test images were required to train the neural network predictor but `kmeans++` as a predictor requires no test images), in addition to considerable savings in computation.

For example, the time taken to train the particular neural network we used was approximately $18$ minutes using the optimized PyTorch library (see training details under the "Details Report" section in Huy (2020)). In general, the time can be much longer for more complicated datasets. On the other hand, our unoptimized algorithm implementation which used the labels of a sample run of `kmeans++` was still able to achieve a better clustering than the neural network predictor with $\alpha = 0.01$ in Algorithm 2 in $4.4$ seconds. In conclusion, we can achieve a better clustering by combining a much weaker predictor with our algorithm with the additional benefit of using a more flexible and computationally inexpensive methodology.

We also conducted an experiment where we only use a small fraction of the predictor labels in our algorithm. We select a $p$ fraction of the images, query their labels, and run our algorithm on only these points. We then report the cost of clustering on the entire dataset as $p$ ranges from $1\%$ to $100\%$. Figure 2(b) shows the percentage increase in clustering cost relative to querying the whole dataset is quite low for moderate values of $p$ but increasingly worse as $p$ becomes smaller. This suggests that the quality of our algorithm does not suffer drastically by querying a smaller fraction of the dataset.

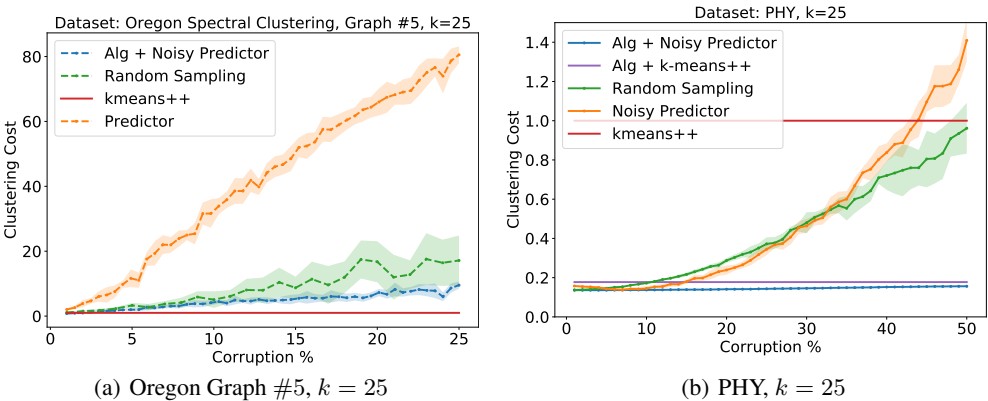

(a) Oregon Graph #5, $k = 25$         (b) PHY, $k = 25$

Figure 3: Our algorithm is able to recover a good clustering even for very high levels of noise.

### E.2 SYNTHETIC DATASET

We use a dataset of $10010$ points in $\mathbb{R}^d$ for $d = 10^3$ created using the `kmeans++` lower bound construction presented in Arthur & Vassilvitskii (2007). The dataset consists of $10$ well separated clusters in $\mathbb{R}^d$: let $e_i$ denote the basis vectors. Our dataset is $\{1000e_i\} \cup \{1000e_i + e_j\}$ for all $1 \le i \le 10, 1 \le j \le 1000$.

From the description of the dataset, we can explicitly calculate the optimal clustering and its cost. Our predictor for this dataset was to take the optimal clustering and randomly change each label

with probability $1/2$ to another uniformly random label. Empirically, kmeans++ seeding returned a clustering that had cost at least **1.9x** the optimal clustering. Furthermore, using just the predictor labels naïvely resulted in a clustering with cost up to five orders of magnitude larger than the optimal clustering. In contrast, our algorithm was able to precisely recover the true clustering after processing the predictor outputs. In addition, applying our algorithm using the labels of a sample run of kmeans++ was also able to precisely recover the true clustering.

### E.3 COMPARISON TO LLOYD'S HEURISTIC

We give both theoretical and empiricial justifications for why our algorithms could be superior to blindly following a predictor and then running Lloyd's heuristic.

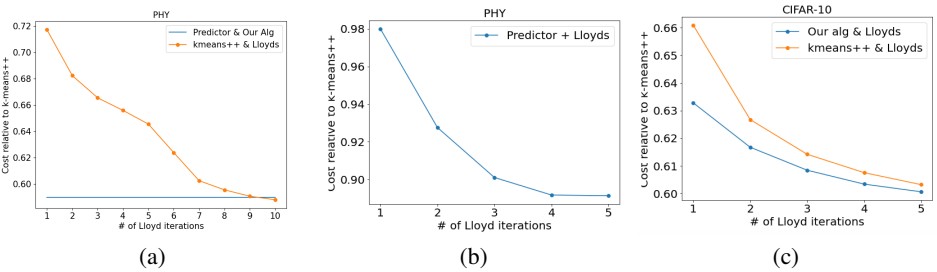

Figure 4: Additional experimental results for comparison to Lloyd's heuristic.

**Empirical Comparison.** We first compared Lloyd's algorithm on kmeans++ seeding to our algorithm with the predictor on the PHY dataset. The predictor is a noisy predictor that has corruption level $50\%$ as described in Section 4 so that outputting the clustering from the predictor alone has cost 1.9x the average kmeans++ cost. Hence, it is clear that the predictor is much worse than kmeans++, yet our algorithm using the predictor (horizontal line in Figure 4(a)) is much better than kmeans++ and Lloyd's algorithm (orange line in Figure 4(a)).

Next, we took the noisy predictor for the PHY dataset with corruption level $50\%$ and repeatedly applied Lloyd's algorithm. We observe that even with $\sim 5$ Lloyd's iterations (Figure 4(b)), the clustering cost does not seem to improve upon kmeans++, much less the clustering output by simply applying our algorithm to the noisy predictions (Figure 4(a)).

Finally, we compared Lloyd's algorithm on kmeans++ seeding to Lloyd's algorithm on the seeding output by our algorithm (with kmeans++ as the predictor) on the CIFAR-10 dataset, similar to the experiments you suggested by Lattanzi & Sohler (2019). Lloyd's algorithm on the seeding output by our algorithm exhibits superior performance than Lloyd's algorithm on kmeans++ (Figure 4(c)), which is consistent with our previous experiments showing that our algorithm improves upon kmeans++. This further strengthens our claim that our algorithm and methodology with provable worst-case guarantees can be applied in conjunction with heuristics such as Lloyd's that do not have provable worst-case guarantees. Moreover, Figure 4(c) indicates that our approach may be more advantageous than just running kmeans++ with heuristics. Note that a naïve implementation of Lloyd's algorithm is $O(ndk)$ time while our algorithm can be implemented in nearly linear time.

We emphasize that all of our above experiments use a noisy predictor with corruption level $50\%$ as input to our algorithm. Our experiments exhibit even better behavior when a clustering produced by kmeans++ is used for a predictor as input to our algorithm. Combined with our other experiments in Section 4, this gives empirical evidence that there exist many scenarios in which running our algorithm with an erroneous predictor is advantageous.

**Theoretical Comparison.** We now provide an example that demonstrates why blindly following the predictor and then running Lloyd's heuristic would run into issues. We emphasize that it is well-known e.g., see Dasgupta (2003); Har-Peled & Sadri (2005); Arthur & Vassilvitskii (2006); Vattani (2011), that Lloyd's algorithm can take a large number of steps to converge. In particular, Vattani (2011) shows that an exponential number of Lloyd iterations can be required for the algorithm to converge to the optimal solution. Nevertheless, we offer the following concrete answer:

We describe a simple set of points that guarantees Lloyd's algorithm will fail. This is based on the example given in Har-Peled & Sadri (2005) and is also conceptually similar to the example given in 1.1. Consider $4n$ points on the real line $x_1, \ldots, x_{2n}, y_1, \ldots, y_{2n}$ so that $y_1 \leq \ldots \leq y_{2n} \leq A < B \leq x_1 \leq \ldots \leq x_{2n}$ where $B - A$ is large. Suppose $k = 2$ in which case the optimal clustering groups all the $y_i$ points together and all the $x_i$ points together as two separate clusters. Suppose the predictor initially gives label "1" to points $y_1, \ldots, y_n$ and gives label "2" to points $y_{n+1}, \ldots, x_1, \ldots, x_{2n}$, so that the predictor has corruption level $\lambda = 1/2$. Then our algorithm that uses this predictor will get a constant-factor approximation in only one iteration. However, since $B - A$ can be arbitrarily large without affecting the optimal clustering cost, blindly listening to the predictor will give a worse clustering. Furthermore, Theorem 2.1 in Har-Peled & Sadri (2005) implies that Lloyd's will take $\Theta(n)$ iterations to converge if initialized using this predictor. Note that even a single (naïve) iteration of Lloyd's algorithm already uses $O(ndk)$ time while our algorithm only uses $O(nd) + \text{poly}(k, 1/\alpha)$ time. Note that there are also more complex examples in higher-dimensional spaces in the literature which provably have even worse convergence rates for Lloyd's method.

### E.4 Conclusion

Although $1.07$-approximation for $k$-means clustering in polynomial time is NP-hard and a clustering consistent with the labels of any predictor with nonzero error can be arbitrarily bad, we give a $(1 + \alpha)$-approximation algorithm that uses the labels output by the predictor as "advice" and runs in nearly linear time. We use a linear number of queries to the predictor, which can be improved to nearly optimal under natural assumptions about the cluster sizes. Our results are well-supported by empirical evaluations and are an important step in demonstrating the power of learning-augmented algorithms beyond the limitations of classical algorithms for clustering-based applications.

## F   Learnability Results

In this section, we present formal learning bounds for learning a good predictor for the learning-augmented $k$-means problem. Namely, we show that a good predictor can be learned efficiently if the problem instances are drawn from a particular distribution. Our result is inspired by derived from data-driven algorithm design and utilizes the PAC learning framework. Formally, our setting is the following.

Suppose there exists an underlying distribution $\mathcal{D}$ that generates independent $k$-means clustering instances, representing the case where similar instances of the $k$-means clustering problem are being solved. Note that this setting also mirrors some of our experiments in Section 4, specifically in the case of our spectral clustering datasets which are derived from snapshots of a dynamic graph across time.

Our goal is to efficiently learn a good predictor $f$ among some family of functions $\mathcal{F}$. The input to each predictor $f$ is a clustering instance $C$ and the output is a feature vector. We assume that the each input instance $C$ is encoded as a vector in $\mathbb{R}^{nd}$, that is, we think of $nd$ as the size of the description of the input. We also assume that the output of $f$ is in $n$ dimensions, which represents the prediction of the oracle. For example in the case of $k$-means clustering, the output can be an integer in $[k]$ for each point in $C$.

To select the "best" $f$, we need to formally define what we mean by best. In most learning settings, a loss function is used to measure the quality of a solution. Indeed, suppose we have a loss function $L : f \times C \to \mathbb{R}$ which represents how well a predictor $f$ performs on some input $C$. For example, $f$ can represent the cluster labels of a points in $C$ and $L$ outputs the cost of the clustering. Alternatively, $L$ can represent an algorithm which uses the hints given by $f$ and performs a clustering algorithm such as some number of steps of Lloyd's heuristic.

Our goal is to learn the best function $f \in \mathcal{F}$ that minimizes the following objective:

$$\mathbb{E}_{C \sim D}[L(f, C)]. \tag{2}$$

We define $f^*$ to be an optimal function $f \in \mathcal{F}$, so that $f^* = \arg\min \mathbb{E}_{C \sim D}[L(f, C)]$. We also assume that for each clustering instance $C$ and each $f \in F$, $f(C)$ and $L(f, C)$ can be computed in time $T(n, d)$ that should be interpreted as a (small) polynomial in $n$ and $d$.

Our main result is the following.

**Theorem F.1.** *There exists an algorithm that uses* $\text{poly}(T(n,d), 1/\varepsilon)$ *samples and returns a* $\hat{f}$ *that satisfies*

$$\mathbb{E}_{C \sim D}[L(\hat{f}, C)] \leq \mathbb{E}_{C \sim D}[L(f^*, C)] + \varepsilon$$

*with probability at least* $9/10$.

The above theorem is a PAC-style bound that shows only a small number of samples are needed in order to have a good probability of learning an approximately-optimal function $\hat{f}$. The algorithm to compute $\hat{f}$ is the following: we simply minimize the empirical loss after an appropriate number of samples are drawn, i.e., we perform empirical risk minimization. This result is proven by Theorem F.3. Before introducing it, we need to define the concept of pseudo-dimension for a function class, which is the more familiar VC dimension, generalized to real functions.

**Definition F.2** (Pseudo-Dimension, e.g., Definition 9 in Lucic et al. (2018))**.** *Let* $\mathcal{X}$ *be a ground set and* $\mathcal{F}$ *be a set of functions from* $\mathcal{X}$ *to the interval* $[0, 1]$. *Fix a set* $S = \{x_1, \cdots, x_n\} \subset \mathcal{X}$, *a set of reals numbers* $R = \{r_1, \cdots, r_n\}$ *with* $r_i \in [0, 1]$ *and a function* $f \in \mathcal{F}$. *The set* $S_f = \{x_i \in S \mid f(x_i) \geq r_i\}$ *is called the induced subset of* $S$ *formed by* $f$ *and* $R$. *The set* $S$ *with associated values* $R$ *is shattered by* $\mathcal{F}$ *if* $|\{S_f \mid f \in \mathcal{F}\}| = 2^n$. *The* pseudo-dimension *of* $\mathcal{F}$ *is the cardinality of the largest shattered subset of* $\mathcal{X}$ *(or* $\infty$).

The following theorem relates the performance of empirical risk minimization and the number of samples needed, to pseudo-dimension. We specialize the theorem statement to our situation at hand. For notational simplicity, we define $\mathcal{G}$ be the class of functions in $\mathcal{F}$ composed with $L$:

$$\mathcal{G} := \{L \circ f : f \in \mathcal{F}\}.$$

Furthermore, by normalizing, we can assume that the range of $L$ is equal to $[0, 1]$.

**Theorem F.3** (Anthony & Bartlett (1999))**.** *Let* $\mathcal{D}$ *be a distribution over problem instances* $C$ *and* $\mathcal{G}$ *be a class of functions* $g : \mathcal{C} \to [0, 1]$ *with pseudo-dimension* $d_{\mathcal{G}}$. *Consider* $t$ *i.i.d. samples* $C_1, C_2, \ldots, C_t$ *from* $\mathcal{D}$. *There is a universal constant* $c_0$, *such that for any* $\varepsilon > 0$, *if* $t \geq c_0 \cdot d_{\mathcal{G}}/\varepsilon^2$, *then we have*

$$\left| \frac{1}{t} \sum_{i=1}^{t} g(C_i) - \mathbb{E}_{C \sim \mathcal{D}} \, g(C) \right| \leq \varepsilon$$

*for all* $g \in \mathcal{G}$ *with probability at least* $9/10$.

The following corollary follows from the triangle inequality.

**Corollary F.4.** *Consider a set of* $t$ *independent samples* $C_1, \ldots, C_t$ *from* $\mathcal{D}$ *and let* $\hat{g}$ *be a function in* $\mathcal{G}$ *that minimizes* $\frac{1}{t} \sum_{i=1}^{t} g(C_i)$. *If the number of samples* $t$ *is chosen as in Theorem F.3, then*

$$\mathbb{E}_{C \sim D}[\hat{g}(C)] \leq \mathbb{E}_{C \sim D}[g^*(C)] + 2\varepsilon$$

*holds with probability at least* $9/10$.

Therefore, the main challenge at hand is to bound the pseudo-dimension of our given function class $\mathcal{G}$. To do so, we first relate the pseudo-dimension to the VC dimension of a related class of threshold functions. This relationship has been fruitful in obtaining learning bounds in a variety of works such as Lucic et al. (2018); Izzo et al. (2021).

**Lemma F.5** (Pseudo-dimension to VC dimension, Lemma 10 in Lucic et al. (2018))**.** *For any* $g \in \mathcal{G}$, *let* $B_g$ *be the indicator function of the region on or below the graph of* $g$, *i.e.,* $B_g(x, y) = sgn(g(x) - y)$. *The pseudo-dimension of* $\mathcal{G}$ *is equivalent to the VC-dimension of the subgraph class* $B_{\mathcal{G}} = \{B_g \mid g \in \mathcal{G}\}$.

Finally, the following theorem relates the VC dimension of a given function class to its computational complexity, i.e., the time complexity of computing a function in the class.

**Lemma F.6** (Theorem 8.14 in Anthony & Bartlett (1999))**.** *Let* $h : \mathbb{R}^a \times \mathbb{R}^b \to \{0, 1\}$, *determining the class*

$$\mathcal{H} = \{x \to h(\theta, x) : \theta \in \mathbb{R}^a\}.$$

*Suppose that any* $h$ *can be computed by an algorithm that takes as input the pair* $(\theta, x) \in \mathbb{R}^a \times \mathbb{R}^b$ *and returns* $h(\theta, x)$ *after no more than* $t$ *of the following operations:*

- *arithmetic operations $+, -, \times$, and $/$ on real numbers,*
- *jumps conditioned on $>, \geq, <, \leq, =$, and $=$ comparisons of real numbers, and*
- *output $0, 1$,*

*then the VC dimension of $\mathcal{H}$ is $O(a^2 t^2 + t^2 a \log a)$.*

Combining the previous results allows us prove Theorem F.1. At a high level, we are instantiating Lemma F.6 with the complexity of *computing* any function in the function class $\mathcal{G}$.

*Proof of Theorem F.1.* First by Theorem F.3 and Corollary F.4, it suffices to bound the pseudo-dimension of the class $\mathcal{G} = L \circ \mathcal{F}$. Then from Lemmas F.5, the pseudo-dimension of $\mathcal{G}$ is the VC dimension of threshold functions defined by $\mathcal{G}$. Finally from Lemma F.6, the VC dimension of the appropriate class of threshold functions is polynomial in the complexity of computing a member of the function class. In other words, Lemma F.6 tells us that the VC dimension of $B_{\mathcal{G}}$ defined in Lemma F.5 is polynomial in the number of arithmetic operations needed to compute the threshold function associated to some $g \in \mathcal{G}$. By our definition, this quantity is polynomial in $T(n, d)$. Hence, the pseudo-dimension of $\mathcal{G}$ is also polynomial in $T(n, d)$ and our desired result follows. □

Note that we can consider initializing Theorem F.1 with specific predictions. If each function in the family of oracles we are interested can be computed efficiently, which is the case of the predictors we employ in our experiments, then Theorem F.1 assures us that we only require polynomially many samples to be able to learn a nearly optimal oracle.

Note that our result is in similar in spirit to the recent paper Dinitz et al. (2021). They derive sample complexity learning bounds for a different algorithmic problem of computing matchings in a graph. Since they specialize their analysis to a specific function class and loss function, their bounds are possibly tighter rather than the possibly loose polynomial bounds we have stated. However, our analysis above is more general as it allows for a variety of predictors, as we employ in our experiments, and loss functions to measure the quality of the predictions.

