# OpenReview forum: "Learning-Augmented $k$-means Clustering"
_ICLR.cc/2022/Conference — ICLR 2022 Spotlight_

### Official Review · Reviewer_dm82 · 2021-10-23

**Correctness:** 4
**Technical Novelty And Significance:** 2
**Empirical Novelty And Significance:** 2
**Recommendation:** 6
**Confidence:** 4

**Main Review:**

The paper is nicely written, with the motivation clearly stated. The $k$-means clustering is a fundamental, yet important problem. The authors consider a setting in which some prior knowledge about the clustering is available, and derive a statistical method to recover the solution with some provable guarantees.

Weakness

• The theorem applies when the label error is small, less than 1/7. However, it might be non-trivial to obtain a predictor with that quality in the first place. For example, in the experiments, the initial solution are derived from $k$-means (Lloyd's) algorithm, which might require many initial seeds to attain a good solution. Are there any guarantees can be made when the initial label error is larger?

• When $k$ gets larger, the $k$-means algorithm (even with $k$-means++) solution can be stuck at local minima, with arbitrarily worse objective [1]. How would algo+$k$-means++/predictor behave compare with $k$-means++ (with multiple seeds)? Can the algorithm help escape the local minima and attain a much better solution? There is a collection of synthetic datasets [1] for k-means to understand the performance of the algorithm. I suggest the authors take these benchmark datasets into consideration for the experiments for evaluation. In the current experiment, only $k=10$ and $k=25$ are tested, and it is hard to see the comparison of algorithms when $k$ gets larger, which is a more challenging case for the $k$-means problem.

• Minor suggestion: the average of $k$-means objectives with multiple seeds are used as a baseline, I think the minimal $k$-means objective over multiple seeds is more reasonable.


[1] Jin, Chi, et al. "Local maxima in the likelihood of gaussian mixture models: Structural results and algorithmic consequences." Advances in neural information processing systems 29 (2016): 4116-4124.
[2] Fränti, Pasi, and Sami Sieranoja. "K-means properties on six clustering benchmark datasets." Applied Intelligence 48.12 (2018): 4743-4759.

**Summary Of The Paper:**

This work aims to find a solution of the k-means clustering problem based on (prior) predicted labels. Here the predicted labels could be obtained from some clustering algorithms or some supervised models, with additional noise. A polynomial time procedure (Algorithm 1) is proposed as follows: for each fitted component from the predicted labels, a robust mean is computed (in a coordinate wise way). These robust means are output as the final clustering solution. The running time is $O(knd\log n)$. The paper rigorously establishes a theoretical guarantee on the approximation ratio of the solution assuming the predictor label has a bounded label error. To further improve the running time, the authors utilize dimension reduction technique to cluster $O(k/\alpha)$ points in dimension $O(\log n)$ and then obtain the label for original data points with approximate nearest neighbor data structure. This modified approach (Algorithm 3) has running time $O(nd\log n+\text{poly}(k,\log n))$ and attains a solution with a similar theoretical guarantee. To empirically evaluate the proposed method, the authors perform experiments on synthetic data and a few real datasets. The experiments demonstrate that the proposed method with $k$-means++ initialization achieves better performance than $k$-means++; moreover, the performance is competitive and robust even when the predictor labels are corrupted.

**Summary Of The Review:**

The paper provides a simple and practical approach to obtain a clustering solution from some predictor labels. A rigorous analysis of the algorithm has been provided. I would recommend the paper to be accepted. For further improvement, the authors need to address, at least empirically, if not theoretically (1) robustness of the algorithm to different level of label error, i.e., the performance of the algorithm when the assumption fails (2) robustness of the algorithm to different $k$, the number of components.

---

### Official Review · Reviewer_Ftwn · 2021-10-31

**Correctness:** 3
**Technical Novelty And Significance:** 4
**Empirical Novelty And Significance:** 4
**Recommendation:** 8
**Confidence:** 5

**Main Review:**

I’d like to point out that this setting/algorithm is conceptually different (and weaker) from many previous papers on learning-augmented algorithms (cf. [Competitive Caching with Machine Learned Advice, Lykouris and Vassilvitskii, J. ACM]), in the sense that this algorithm does not have strong guarantee when the prediction is very inaccurate (for instance, Theorem 2.1 requires \alpha to be at most 1/ 7). Instead, the focus of this algorithm seems to be “de-noising” an already good but slightly noisy predictor. This could be a limitation of the work, since “good” predictors themselves may not be easily obtained, and it could be that the major issue of using predictions is not to be “misled”. Nonetheless, I can still see that this “de-noising” setting make sense, and it is relevant in dealing with adversarial data set. But I’d still like to see how your algorithm performs, when the predictor is inaccurate (e.g., it is only 10 approximation?)

I also have other concerns about some proof details and the experiment results. The major ones are listed as follows.
1.	Lemma A.4. In the proof, one of the step is to use Chernoff bound to argue |I \cap X_2| >= m(1 – 6 \alpha). Can you elaborate how Chernoff bound is applied? Note that I and X_2 are not independent and both of them are random (recalling that X_2 is the complement of X_1, and I is defined w.r.t. X_1, so they both depend on the random set X_1).
2.	In Figure 1 (a), it seems Alg+Predictor and Alg+k-means++ are highly correlated, but the performance of predictor and k-means++ are shown to be quite different in some of the graphs. Can you explain why this happens? Also, this correlation is not observed in Figure 1 (b). Can you also explain why is the difference?
3.	In your experiments, it seems k-means++ baseline only runs the random seeding step, without any iteration of Lloyd heuristic. I suggest to run at least one round of Lloyd’s heuristic after your seeding as another baseline, since your Algorithm 1 is essentially Lloyd if in Algorithm 2 no bad point is eliminated.

Overall, this is an interesting paper, and it is somewhat the first of its kind, in the sense that it studies how machine-learned advice could help to improve the *time complexity* of k-clustering, instead of looking at the competitive ratio as in the commonly studied online setting. I would be glad to accept the paper if the authors could properly address my major concerns in the follow-up discussions.

Minor comments:

1.	Lemma A.9, please specify what “these points” refer to, in the second line of the statement.
2.	In the proof of Lemma A.11, could you remind the definition of p, in the displaymath?
3.	Also in the proof of A.11, 2. The equation below ‘By Markov’s Inequality, …’, should be ‘\sum_{i \in [k]} |X_i|*||C_i-\gamma_i||_2^2’ instead of ‘\sum_{i \in [k]} ||C_i-\gamma_i||_2^2’ ?
4.	Page 9, the paragraph about CIFAR-10. You mentioned that your algorithm “could improve upon this highly precise predictor” – I think the claim is a bit weird, because the neural network predictor is for the purpose of classification, while your task is clustering. The objectives might be related, but I don’t see a strong correlation, and it’s possible that this neural network baseline has a bad clustering cost. Hence, your improvement over the neural network one is not clearly convincing to me. I suggest to also show the clustering cost of the neural network predictor.
5.	I don’t get why your Theorem 2.1 needs O(kdn) time – it seems to be Algorithm 1 + Algorithm 2 only takes O(nd) time, because each iteration of the for-loop in Algorithm 1 takes O(|Y_i|) time/accesses to the predictor, and \sum_i |Y_i| = n? This also confuses me about the necessity/improvement of Theorem 3.4.


**Summary Of The Paper:**

The paper studies k-means problem in a learning-augmented setting. Recall that in k-means, a point set $P \subset \mathbb{R}^d$ is given and the goal is to find a set $C$ of $k$ centers, such that $\mathrm{cost}(P,C) = \sum_{p \in P} \min_{c \in C} \|p-c\|_2^2$ is minimized. In addition to the input point set, a predicted solution, which is an approximately optimal clustering of $P$, is also provided in the proposed learning-augmented setting. The algorithm can access this solution by querying the predictor the cluster that a point $x$ belongs to.

The main result is an algorithm that can leverage the predictions. For $\alpha \in (10 \log n / \sqrt{n}, 1/7)$, given access to a predictor with label error rate $\lambda \leq \alpha$, and let $\gamma \geq 1$ be a sufficiently large constant, if the predicted solution satisfies all clusters has at least $\gamma k \log k / \alpha$ points, then the algorithm outputs a $(1+20\alpha)$-approximate solution with probability at least $3/4$, using $O(nd \log n + \mathrm{poly}(k, \log n))$ total time.  Furthermore, the algorithm only uses $\tilde{O}(k / \alpha)$ queries to the predictor for outputting the centers. The authors claim that for any $\delta \in (0,1]$, any algorithm makes $O(k^(1-\delta)/(\alpha \log n))$ queries to the predictor with label rate $\alpha$ cannot output a $(1+C \alpha)$-approximate solution for k-means problem in $2^{O(n^(1-\delta))}$ time, assuming ETH.  The authors also experimented with their algorithm on three datasets, along with three different predictors. The experiment result shows that their algorithm significantly improves the performance when using with predictor or k-means++, and has a stable accuracy against corrupted predictors.

Technically, since it is guaranteed that the predicted solution is a $(1+\alpha)$ approximately optimal solution, with some of the labels corrupted, Algorithm 1 aims to reconstruct centers of each cluster $X$ in $(1+\alpha)$-approximately optimal solution, i.e. find point $c$ such that $\mathrm{cost}(X, c) \leq (1+O(\alpha)) \mathrm{cost}(X, C_X)$. Notice that $d$ dimensions are independent of each other in k-means objective. Hence Algorithm 1 can estimate each coordinate of $c$ independently, which is implemented by Algorithm 2.  Then Algorithm 2 (randomly) divides the input (1D) data points into two parts, and one part is used for the computation of interval $I$, an estimation of the value range of the uncorrupted data points. Then the points outside $I$ are filtered out and the mean of the remaining coordinates are returned as the final estimation.


**Summary Of The Review:**

This is an interesting paper that has both conceptual and technical novelties. However, I also see some technical issues in the paper, and I'd like to accept the paper if those issues are addressed properly.

---

### Official Review · Reviewer_4io3 · 2021-11-02

**Correctness:** 4
**Technical Novelty And Significance:** 2
**Empirical Novelty And Significance:** 2
**Recommendation:** 8
**Confidence:** 3

**Main Review:**

Strengths:

1. The problem is interesting. Given the fact that hard clustering is generally NP-hard in the worst-case, it make sense to try and mimic/find scenarios where this discouraging fact can be bypassed, e.g., by introducing reasonable side information. As the authors mentioned, perhaps the first attempt for introducing such side information is the SSAC framework. This paper introduce another possible approach.

2. The paper is well-written and motivated.

3. The numerical part is thorough.

Weaknesses:

1. Similarly to the SSAC framework, it is unclear to me if such a predicator(s) exist in practice. I appreciate that, at least, based on the way things are presented the current framework seems to be more "practical" than the SSAC framework. In particular, I am not completely sure what is required by the predictor; unless I am missing something trivial, we need the \alpha to be less than 1/7, which to me seems completely not easy to obtain. How you managed to do that in your experiments?

2. I am not sure if it is a real weakness, but I find both the algorithm and (especially) the analysis quite standard or at least not surprising. Perhaps the authors could elaborate a bit more on the technical novelty in their proofs.

**Summary Of The Paper:**

This paper considers the problem of k-means clustering with the aid of a predictor which supplies a proxy to the optimal clustering subject to some possible errors. The motivation for this setting is the inherent computational issues with solving the vanilla k-means clustering problem. For this model, the authors propose and analyze an efficient algorithm whose approximation factor scales gracefully with the predictor error guarantees.

**Summary Of The Review:**

See above.

---

### Author Response · Authors · 2021-11-12
**Rebuttal Revision and Learnability Theory**

Dear all reviewers,

We've uploaded a rebuttal revision that incorporates the individual responses to each reviewer. Moreover, we included a section in Appendix F of the revision that formally describes conditions under which it would be theoretically possible to provably learn a good predictor. We hope that in addition to the good performance of our experimental results, this theory confirms to all reviewers that such predictors actually do exist in practice.

---

### Decision · Program_Chairs · 2022-01-20

**Decision:**

Accept (Spotlight)

**Comment:**

One might assume that the k-means problem has already been beaten to
death, but this paper shows there are still remaining questions. And
rather interesting ones at that, with a novel angle of having
additional help from a prediction algorithm of cluster
memberships. This connects to learning-augmented algorithms research.

The reviewers agreed that the problem is interesting and gives a novel
angle, and the interestingness stems from novelty, and the ability to
"escape" from NP-hardness.

The reviewers and authors had nice discussions about details and
conclusions, on how limiting is it that the authors focus on
reasonably accurate predictors, for instance, and where could the
predictors come from. This is a good paper, and hopefully the
discussion helped make it even better.